# Single-cell transcriptional profiling of human thymic stroma uncovers novel cellular heterogeneity in the thymic medulla

Jhoanne L. Bautista[1,2], Nathan T. Cramer[1], Corey N. Miller[1], Jessica Chavez[1], David I. Berrios[1], Lauren E. Byrnes [1], Joe Germino [1,3,4,5], Vasilis Ntranos[1,3,4,5], Julie B. Sneddon [1,6,7,8], Trevor D. Burt [6,9,10], James M. Gardner[1,2], Chun J. Ye [4,11,12,13], Mark S. Anderson[1,14] & Audrey V. Parent [1,14 ✉]

The thymus' key function in the immune system is to provide the necessary environment for the development of diverse and self-tolerant T lymphocytes. While recent evidence suggests that the thymic stroma is comprised of more functionally distinct subpopulations than previously appreciated, the extent of this cellular heterogeneity in the human thymus is not well understood. Here we use single-cell RNA sequencing to comprehensively profile the human thymic stroma across multiple stages of life. Mesenchyme, pericytes and endothelial cells are identified as potential key regulators of thymic epithelial cell differentiation and thymocyte migration. In-depth analyses of epithelial cells reveal the presence of ionocytes as a medullary population, while the expression of tissue-specific antigens is mapped to different subsets of epithelial cells. This work thus provides important insight on how the diversity of thymic cells is established, and how this heterogeneity contributes to the induction of immune tolerance in humans.

[1] Diabetes Center, Department of Medicine, University of California, San Francisco, San Francisco, CA, USA. [2] Department of Surgery, University of California, San Francisco, San Francisco, CA, USA. [3] Department of Epidemiology & Biostatistics, University of California, San Francisco, San Francisco, CA, USA. [4] Bakar Institute for Computational Health Sciences, University of California, San Francisco, San Francisco, CA, USA. [5] Department of Bioengineering and Therapeutic Sciences, University of California, San Francisco, San Francisco, CA, USA. [6] Broad Center of Regeneration Medicine and Stem Cell Research, University of California, San Francisco, San Francisco, CA, USA. [7] Department of Anatomy, University of California, San Francisco, San Francisco, CA, USA. [8] Department of Cell and Tissue Biology, School of Dentistry, University of California, San Francisco, San Francisco, CA, USA. [9] Division of Neonatology, Department of Pediatrics, University of California, San Francisco, San Francisco, CA, USA. [10] Division of Neonatology and the Children's Health & Discovery Initiative, Department of Pediatrics, Duke University School of Medicine, Durham, NC, USA. [11] Institute for Human Genetics, University of California, San Francisco, San Francisco, CA, USA. [12] Division of Rheumatology, Department of Medicine, University of California, San Francisco, San Francisco, CA, USA. [13] Chan Zuckerberg Biohub, San Francisco, CA, USA. [14] These authors contributed equally: Mark S. Anderson, Audrey V. Parent. ✉email: Audrey.Parent@ucsf.edu

The thymus is a critical primary lymphoid organ that supports the development of a diverse, self-tolerant peripheral T cell pool. The predominant stromal cells found in the postnatal thymus are thymic epithelial cells (TECs), mesenchyme, endothelium, and non-lymphoid hematopoietic cells (dendritic cells and macrophages). Cortical TECs (cTECs) are responsible for T lineage commitment and positive selection of early thymocytes, whereas medullary TECs (mTECs) participate in the deletion of autoreactive cells and the final stages of thymocyte maturation. While the function of TECs has been well established, their developmental origins as well as the mechanisms controlling their maintenance are still not clear. Evidence from mouse models has revealed the existence of common bipotent thymic epithelial progenitor cells (TEPCs) with cTEC-like properties that have the potential to give rise to both cTECs and mTECs in the fetal and early postnatal thymus[1–5]. In contrast, generation of TECs in adult murine thymus likely depends on lineage-restricted progenitors as bipotent TEPCs have been shown to become quiescent[6]. It is however unknown how these observations translate to human thymic development where evidence of bipotent TEPCs is lacking and knowledge of postnatal tissue regulation is minimal.

Recent studies have shown that the thymic medulla is more heterogeneous than previously appreciated. For example, in addition to the well-characterized population of mature mTECs expressing AIRE and tissue-specific antigens (TSAs), the murine thymic medulla contains functionally diverse populations of CC-chemokine ligand 21 (CCL21)-expressing mTECs, corneocyte-like mTECs (also known as post-AIRE mTECs), and thymic tuft cells[7–10]. It has been shown that CCL21-producing cells are critical for the recruitment of positively selected CCR7+ thymocytes into the medulla[11], whereas thymic tuft cells help shape thymocyte development by promoting an IL-4-enriched environment[8]. Therefore, while the contributions of such subsets to thymocyte development is an active area of investigation, there is mounting evidence that the murine TEC compartment is highly diverse and functionally compartmentalized. A recent study also noted that the human thymus contains many distinct subpopulations[12] but a thorough characterization of the stromal compartment was not the primary focus of this work.

Here we use single-cell RNA sequencing (scRNA-seq) to investigate cellular heterogeneity in the human thymic microenvironment at different time points during development. Using these data, we identify candidate pathways that regulate TEC fate commitment and uncover previously uncharacterized TEC markers. Importantly, we confirm the presence of distinct mTEC subsets (AIRE+, corneocyte-like, CCL21+, and tuft), identify ionocytes as a subset of epithelial cells present in the human thymic medulla, and provide the first transcriptomic characterization of rare populations of thymic ciliated cells and neuronal cells. Finally, we analyze the expression of disease-relevant genes in the epithelial compartment in an effort to better understand how immune tolerance is established in the human thymus. Together, this work advances our understanding of human TEC development and provides an unbiased resource to study human TEC heterogeneity and its relevance to human autoimmune diseases.

## Results

**Single-cell profiling of stromal cells from human thymus**. To identify the different cell types comprising the human thymic microenvironment, we performed scRNA-seq of stromal cells isolated from fetal, postnatal, and adult tissue. Stromal cells were obtained by enzymatic digestion of thymic tissue followed by depletion of CD45-positive immune cells using magnetic beads or fluorescence-activated cell sorting (FACS)-based purification of CD45 negative cells (Fig. 1a). These procedures led to the enrichment of both EpCAM$^+$ CD45$^-$ epithelial cells and EpCAM$^-$ CD45$^-$ non-epithelial stromal cells (Fig. 1b and Supplementary Fig. 1a). Cells isolated from two fetal (19 and 23 gestational weeks), two postnatal (6 days old and 10 months old), and one adult (25 years old) samples were analyzed (Fig. 1c). Following filtering and batch correction using BBKNN[13], our final dataset contained 68,008 cells, which were taken forward for further analysis. Employing an unsupervised graph-based clustering strategy, we identified 12 stromal clusters (Fig. 1d). Known markers were used to identify three epithelial (EPCAM and KRT8 as general epithelial markers and FOXN1, PSMB11, LY75, CLDN4, AIRE, IVL, NEUROD1, MYOD1 as markers of specific subsets), one mesenchymal (PDGFRA, LUM, LAMA2), one pericyte (PDGFRB, MCAM, CSPG4 also known as NG2), one vascular arterial endothelial (PECAM1, VEGFC, GJA4), two vascular venous endothelial (PECAM1, ACKR1, SELE, APLNR), one lymphatic endothelial (LYVE1, PROX1, CCL21), one red blood cell (GYPA, HBA1, HBG1), one immune cell (PTPRC, CD3D, CD7), and one mesothelial (MSLN, UPK3B, PRG4) sub-clusters (Fig. 1e, Supplementary Fig. 1b–d and Supplementary Data 1).

**Stromal cells provide complementary factors critical for TEC development and thymocyte migration**. Studies in animal models have shown that neural crest, mesenchyme, and endothelial cells are important for the establishment of a thymic microenvironment that supports thymopoiesis through the production of soluble factors and cell–cell interactions[14–18]. The function and cell-type specificity of these soluble factors in human thymic development are, however, not well understood. To define the signals provided by human stromal cells, we analyzed the expression of soluble factors and receptors of the WNT, BMP, transforming growth factor beta (TGF beta), insulin-like growth factor (IGF), and fibroblast growth factor (FGF) signaling pathways, which have been described as critical regulators of the development and function of TECs[6,19–25] (Fig. 1f and Supplementary Fig. 1e, f). Our analysis revealed that mesenchymal cells expressed many ligands and regulators of these critical pathways, including WNT5A, RSPO3, SFRP2, IGF1, and FGF10 (Fig. 1f) while receptors for these factors (ROR1, ROR2, RYK, IGF1R, and FGFR2) were found in epithelial cells (Supplementary Fig. 1e, f). Notably, BMP4, FGF7 (also known as KGF), and the secreted WNT inhibitor Frizzled Related Protein FRZB were expressed more frequently in postnatal and adult mesenchymal cells compared to fetal mesenchyme (Fig. 1g), suggesting that TEC differentiation and proliferation is differentially regulated by mesenchymal factors over time. Most endothelial cells expressed TGFB1 and TGFBR2 while arterial and lymphatic subsets had high levels of chemokines known to promote homing of hematopoietic progenitors to the thymus (CXCL12 or CCL21)[26]. Endothelial cells also expressed extracellular matrix and adhesion molecules such as fibronectin (FN1) and LGALS3 that have been shown to regulate thymocyte migration[27,28] (Fig. 1f). Protein expression of fibronectin in endothelial cells was confirmed by immunofluorescence (Fig. 1h). Epithelial cells and mesothelium were enriched for many WNT ligands while mesothelial cells also expressed WNT signaling modulators (RSPO1, RSPO3, SFRP2, SFRP5) and BMP4 (Fig. 1f). Pericytes expressed FRZB as well as WNT6, BMP5, and FGF7. Interestingly, the gene encoding the subunits of Activin A (INHBA), which was recently shown to be important for TEC differentiation[6], was expressed almost exclusively by pericytes (Fig. 1f, i). In contrast, the activin antagonist follistatin (FST), which promotes TEPC maintenance and inhibits

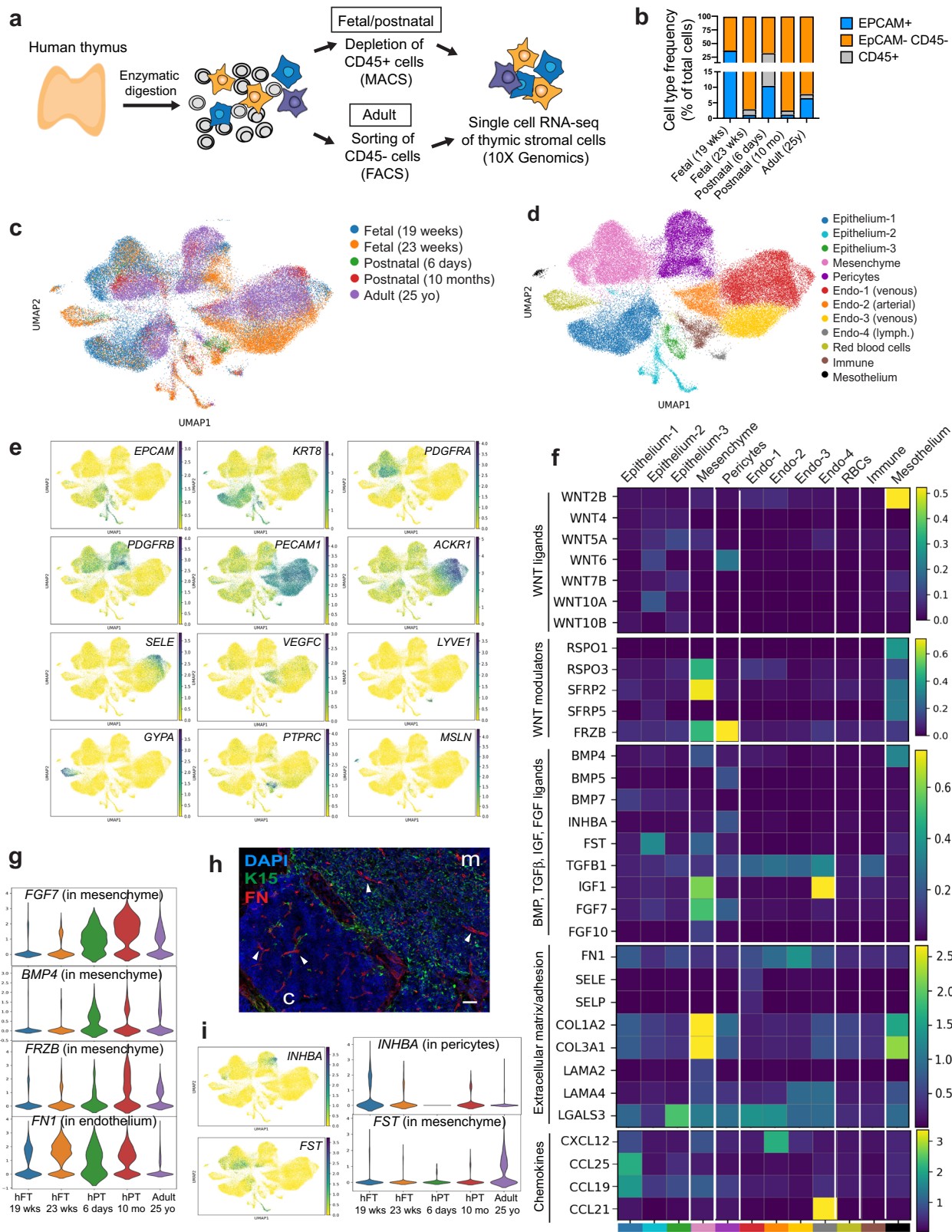

differentiation[6], was found mostly in adult mesenchymal cells and a subset of epithelial cells (Fig. 1f, i). Our analysis thus revealed that human mesenchyme, endothelial cells, and pericytes likely play complementary roles in supporting thymopoiesis by providing factors critical for TEC development or through expression of chemokines and adhesion molecules that regulate migration of hematopoietic progenitors.

**Profiling of human TECs at different stages**. To further define heterogeneity within the epithelial compartment, the three epithelial superclusters (Fig. 1d, Epithelium-1, -2, and -3) were divided into nine distinct sub-clusters (Fig. 2a, b). The clusters were annotated based on a combination of known TEC markers and a list of differentially expressed genes (Fig. 2c, d and Supplementary Data 2). Two clusters expressed genes characteristic

**Fig. 1 Single-cell profiling of stromal cells from human thymus. a** Workflow of tissue preparation for single-cell transcriptome profiling of human thymic stromal cells. CD45-negative cells were enriched using magnetic-activated cell sorting (MACS) or fluorescence-activated cell sorting (FACS). **b** Stacked bar graph of cell-type frequencies in each sample. Source data are provided as a Source Data file. **c** UMAP visualization of thymic stromal cells colored by age group. **d** UMAP visualization of thymic stromal cells colored by cell types. **e** UMAP visualization of the expression of known marker genes used for cell cluster identification. **f** Heatmap showing average expression of soluble factors, extracellular matrix/adhesion molecules, and chemokines in each stromal cluster. **g** Violin plots showing feature gene expression in human fetal thymus (hFT), human postnatal thymus (hPT) or adult thymus. Colors represent age group. **h** Immunofluorescence analysis of fibronectin (FN) and K15+ immature TECs/mTECs (green) expression in human fetal thymus (hFT). Arrows point to blood vessels with high expression of FN in the cortex (c) and medulla (m). Scale bar, 50 μm. Staining was repeated twice with similar results. **i** UMAP and violin plots showing expression of the activin ligand (*INHBA*) and inhibitor (*FST*) in human fetal thymus (hFT), human postnatal thymus (hPT) or adult thymus. Colors represent age group.

of cTECs (*PSMB11*, *PRSS16*, *CCL25*). Cells in the cTEC$^{lo}$ cluster expressed lower levels of functional genes (HLA class II, *PSMB11*, *PRSS16*, *CCL25*) and contained more KI67+-proliferating cells (Fig. 2c, d). Genes characteristic of mTECs were detected in three clusters corresponding to mTEC$^{lo}$ (*CLDN4*, lower levels of HLA class II), mTEC$^{hi}$ (*SPIB*, *AIRE*, *FEZF2*, higher levels of HLA class II), and corneocyte-like mTECs (*KRT1*, *IVL*). Cells in the mTEC$^{lo}$ cluster expressed high levels of the chemokine *CCL21*, reminiscent of the CCL21-expressing mTEC$^{lo}$/jTEC population described in mice[29–32]. We also identified one cluster of cells, marked as immature TEC, which express canonical TEC identity genes (*FOXN1*, *PAX9*, *SIX1*) but lacked functional genes characteristic of cTECs or mTECs. These immature cells were found in all samples (Fig. 2e and Supplementary Fig. 2a) and possibly represent progenitors that are not committed to a specific lineage or cells that have lost their differentiated phenotype over time. Notably, there were very few cTECs or mTEC$^{hi}$ detected in adult thymus (Fig. 2e and Supplementary Fig. 2a), suggesting that there is an accumulation of immature TECs to the detriment of functional TECs in older tissue. This idea is supported by immunofluorescence analysis showing a reduced number of AIRE+ mTEC$^{hi}$ in the medulla (Fig. 2f) and large cortical and medullary areas lacking TECs in older tissue (Fig. 2g). Finally, we found three more clusters that were identified as neuroendocrine (*BEX1*, *NEUROD1*), muscle-like myoid (*MYOD1*, *DES*), and myelin+ epithelial cells (*SOX10*, *MPZ*) (Fig. 2c, d and Supplementary Data 2). The occurrence of neuroendocrine and myoid cells in the human thymus has been noted in previous studies and their transcriptome has recently been published[12,33,34], but these studies did not report the presence of the myelin+ population.

To further validate these findings, we compared our dataset to a human thymus dataset recently published by Park et al.[12]. Both datasets were merged as outlined in the "Methods" section, and clusters were annotated using the TEC markers described in Fig. 2c, d (Supplementary Fig. 2b–f). While most subsets were present in both datasets, one population of cells originating from early fetal samples (<12 weeks) was found only in the Park dataset (early cTEC) (Supplementary Fig. 2b–e). Importantly, the presence of immature TECs as well as the dramatic decline in functional TECs in adult tissues was also observed in the Park dataset (Supplementary Fig. 2g).

**Identification of additional TEC markers.** To gain more insight into the nature of the immature TEC cluster, we sub-clustered the cells to analyze them at a higher resolution (Fig. 3a, b). Two subpopulations were identified (immature TEC-1 and immature TEC-2) that expressed distinct markers (Fig. 3c and Supplementary Data 3). Interestingly, the expression of some genes enriched in immature TEC-2 (*IGFBP5*, *NNMT*, *MAOA*, *DPYS*, *FKBP5*, *GLUL*) was markedly higher in adult cells compared to fetal and postnatal tissues (Fig. 3d). Higher expression of these genes could also be detected in adult tissues from the Park dataset, with one sample (19yo) having markedly higher expression than the other

(35yo) (Supplementary Fig. 3a). The atypical cadherin gene *CDH13* was also enriched in immature TECs (Fig. 3e) and its expression in TECs was confirmed at the protein level by immunofluorescence (Fig. 3e). Given the major decline in thymic function with age, these genes represent interesting factors to study in the context of thymic involution.

We next analyzed our dataset in combination with the Park et al. dataset to uncover additional markers of TEC subsets. The zinc-finger protein *ZBED2*, a transcription factor without a murine counterpart that has recently been linked to the maintenance of the basal state in human keratinocytes[35], was identified as a gene highly expressed in immature TECs and cTECs in both datasets (Fig. 4a, Supplementary Fig. 3b and Supplementary Data 2). Genes regulating TGF-β signaling, including *TDGF1* (also known as CRIPTO) and *CTGF* as well as IGF signaling modulators (*IGFBP5* and *IGFBP6*), were also enriched in immature TECs and cTECs. While the expression of many cytokeratins has been extensively studied and used as markers of specific TEC subsets, the expression of *KRT15* has only been reported in stem cell-derived TEPCs[36]. This cytokeratin is particularly interesting since it is found in multipotent progenitor populations in the hair follicle, esophageal epithelium, and small intestine[37–39]. In both datasets, *KRT15* was highly expressed in mTEC$^{lo}$ but was also detected in immature TECs and its expression increased over time (Fig. 4b and Supplementary Fig. 3b). Immunofluorescence confirmed that KRT8+/KRT5+ cells found at the cortico-medullary junction, which potentially mark immature TECs, expressed low level of KRT15 (Fig. 4c, arrows) while KRT15$^{hi}$ cells were found in the medulla and co-expressed KRT5, likely marking CCL21+ mTEC$^{lo}$. Flow cytometric analysis also revealed that most TECs isolated from adult tissue expressed a combination of KRT8, KRT5, and KRT15 (Fig. 4d).

We also identified genes enriched in mTEC$^{lo}$ (*GABRA5*, *LYPD1*), mTECs$^{hi}$ (*CLEC7A*, *MARCO*, *FXYD2*, *FXYD3*, *IL4I1*, *CHI3L1*, *CD70* (also known as CD27L), *TNFRSF9*), or corneocyte-like mTECs (*FXYD3*, *IL1RN*, *LYPD2*) (Fig. 4a). The proneural basic helix–loop–helix (bHLH) transcription factor Achaete-scute complex 1 (*ASCL1*) was enriched in multiple epithelial subsets, including cTEC$^{hi}$, mTEC$^{lo}$, a subset of AIRE+ mTEC$^{hi}$, and neuroendocrine cells (Fig. 4a, e). The role of this chromatin remodeling factor is well-characterized in the brain, where it is expressed in dividing neural progenitors and promotes their proliferation, specification, and differentiation into neurons[40,41]. ASCL1 has also been shown to play a role in the development of neuroendocrine cells in the lung[42]. Expression of ASCL1 in the medulla of fetal and postnatal human thymus as well as in the medulla of murine thymus was confirmed by immunofluorescence, with expression also observed in the cortex of human fetal thymus only (Fig. 4f and Supplementary Fig. 3c). Co-expression of ASCL1 with KRT15 was detected in a subset of medullary cells, likely representing CCL21+ mTEC$^{lo}$ while KRT15-negative cells likely mark AIRE+ mTEC$^{hi}$. Immunofluorescence staining confirmed the presence of cells that co-expressed ASCL1 and AIRE (Fig. 4g).

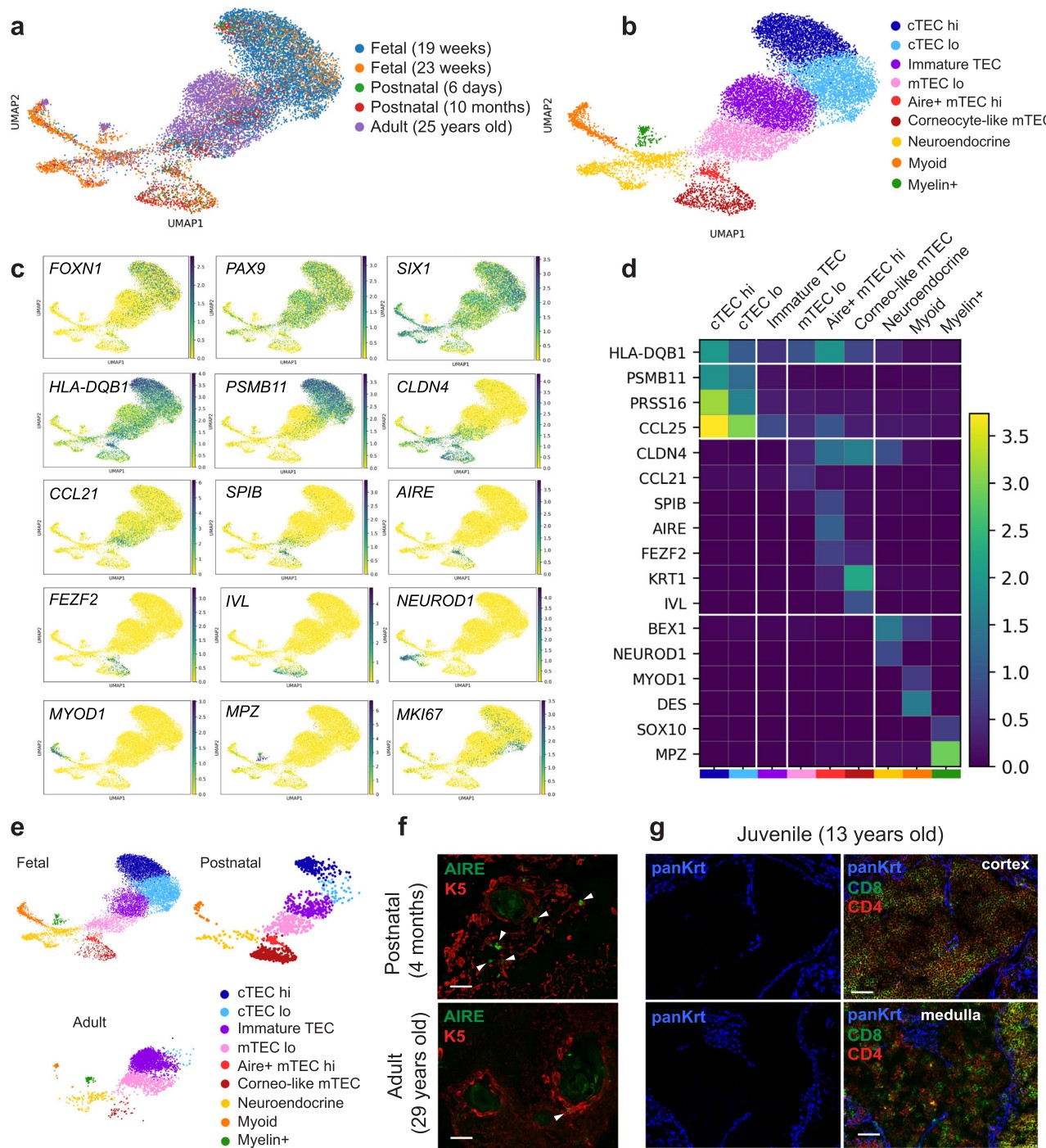

**Fig. 2 Profiling of human thymic epithelial cells at different stages. a** UMAP visualization of epithelial cells colored by age group. **b** UMAP visualization of epithelial cells colored by cell types. **c** UMAP visualization of the expression of known marker genes used for cell cluster identification. **d** Heatmap showing the average expression of known marker genes in each epithelial cluster. **e** UMAP visualization of epithelial subsets in fetal, postnatal, and adult samples. **f** Immunofluorescence staining of AIRE+ (green) K5+ (red) mTEChi cells in postnatal and adult tissues. Arrowheads point to AIRE+ cells. Scale bars, 50 μm. **g** Thymic tissue from a 13-year-old donor was stained with CD8 (green) and CD4 (red) antibodies to visualize thymocytes together with a wide spectrum cytokeratin antibody to identify epithelial cells (blue). Scale bars, 50 μm. Staining in **f** and **g** was repeated twice with many donors with similar results.

Notably, known ASCL1 target genes such as *INSM1*, *DLL3*, *HES6*, *ST3GAL5*, *LYPD1*, and *POU4F1* were detected in TECs (Supplementary Fig. 3d), suggesting activity of ASCL1 rather than expression as part of a promiscuous gene expression program.

Given the established role of ASCL1 in neural progenitors, we wanted to assess whether this transcription factor also marks a pool of progenitor cells in the thymus. We performed an in vivo genetic lineage-tracing experiment using a fluorescent reporter system (Ascl1creERT2; Rosa26CAG-stopflox-tdTomato) (Fig. 4h). In this model, tamoxifen treatment permanently labeled Ascl1-expressing cells and their progeny, allowing us to distinguish between long-lived progenitor cells (labeled cells can be found long after Cre induction) and transit-amplifying cells (reporter-expressing cells diminish over time). These mice were also

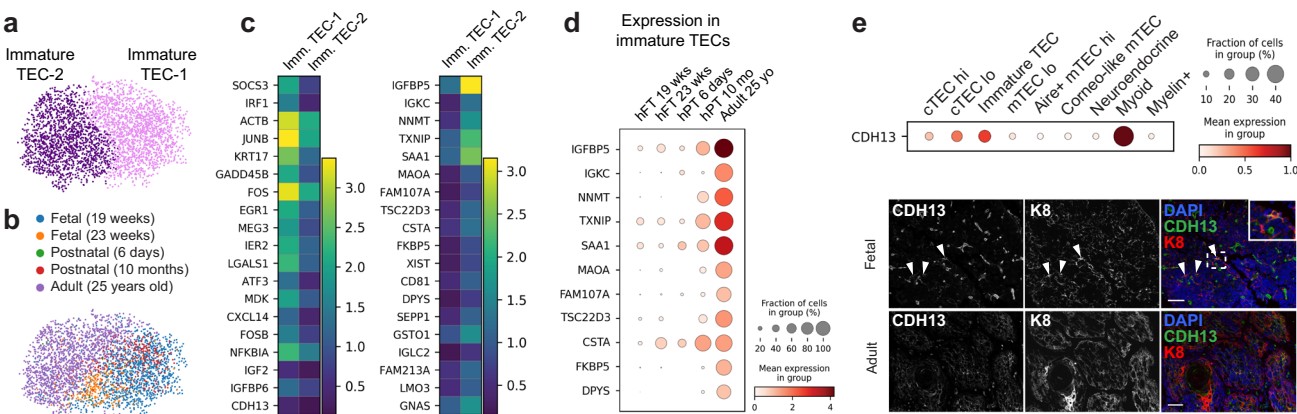

**Fig. 3 Analysis of immature TECs. a, b** UMAP visualization of immature TECs colored by cell type (**a**) or age (**b**). **c** Heatmap showing the expression of marker genes in each immature TEC (imm. TEC) cluster. **d** Dot plot of immature TEC gene expression in human fetal thymus (hFT), human postnatal thymus (hPT), or adult thymus. **e** Expression of CDH13 in epithelial subsets was confirmed by immunofluorescence analysis of human fetal thymus and human adult thymus. Scale bars, 50 μm. Staining was repeated three times with similar results.

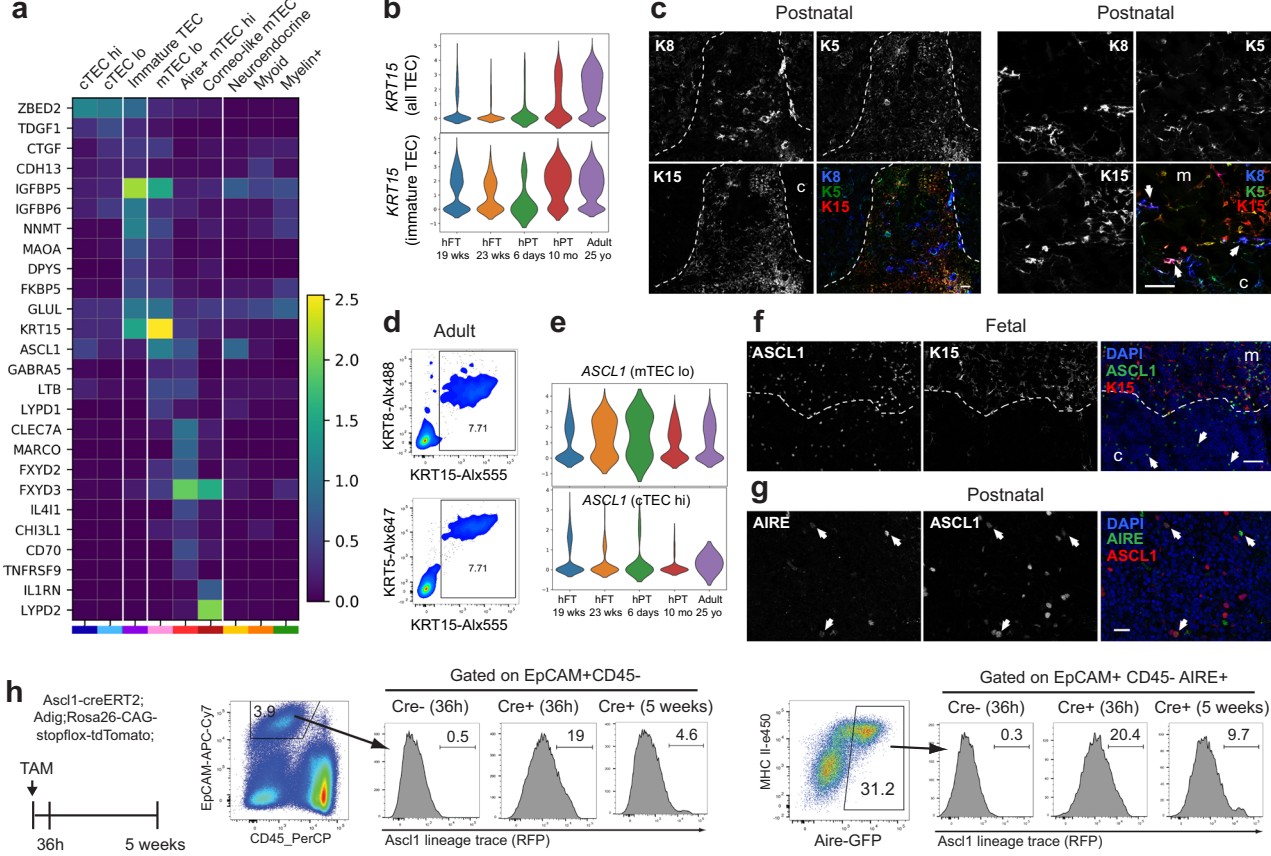

**Fig. 4 Identification of new TEC markers. a** Heatmap showing the expression of newly identified marker genes in each epithelial cluster. **b** Violin plots of KRT15 expression in all TECs and in immature TECs. **c** Immunofluorescence analysis of KRT15 expression in postnatal human thymus. KRT8 (blue) and KRT5 (green) are also included as markers of TECs. Dotted line indicates the separation between cortex (c) and medulla (m). A higher magnification showing that KRT15 is expressed at low levels in KRT8+KRT5+ immature TECs and at higher level in mTECs is shown in the right panels. Medullary area is marked with "m" while cortical area is marked with "c". Arrows point to examples of KRT8+KRT5+ immature TECs. Scale bars, 50 μm. **d** Flow cytometric analysis of KRT15, KRT8, and KRT5 expression in TECs isolated from adult thymus. **e** Violin plots of ASCL1 expression in mTEC lo and cTEC hi.
**f** Immunofluorescence analysis of human fetal thymus demonstrating that ASCL1 (green) is expressed in KRT15$^{hi}$ (red) mTECs. ASCL1 expression is also detected in the cortex of fetal thymus (arrows). Dotted line marks the separation between cortex (c) and medulla (m). Scale bar, 50 μm.
**g** Immunofluorescence analysis of postnatal thymus showing that expression of ASCL1 (red) in the thymic medulla partially overlaps with AIRE (green). Scale bar, 20 μm Arrows point to double positive mTECs. **h** Ascl1-lineage trace (RFP) in TECs at 36 h (*n* = 3 mice) and 5 weeks (*n* = 3 mice) post-tamoxifen (TAM) treatment. Percentage of RFP-labeled Aire+ TECs is shown. Expression of RFP in TECs from Cre-negative mice is also presented as negative control (*n* = 7 mice). Staining in **c**, **f**, and **g** was repeated at least three times with similar results.

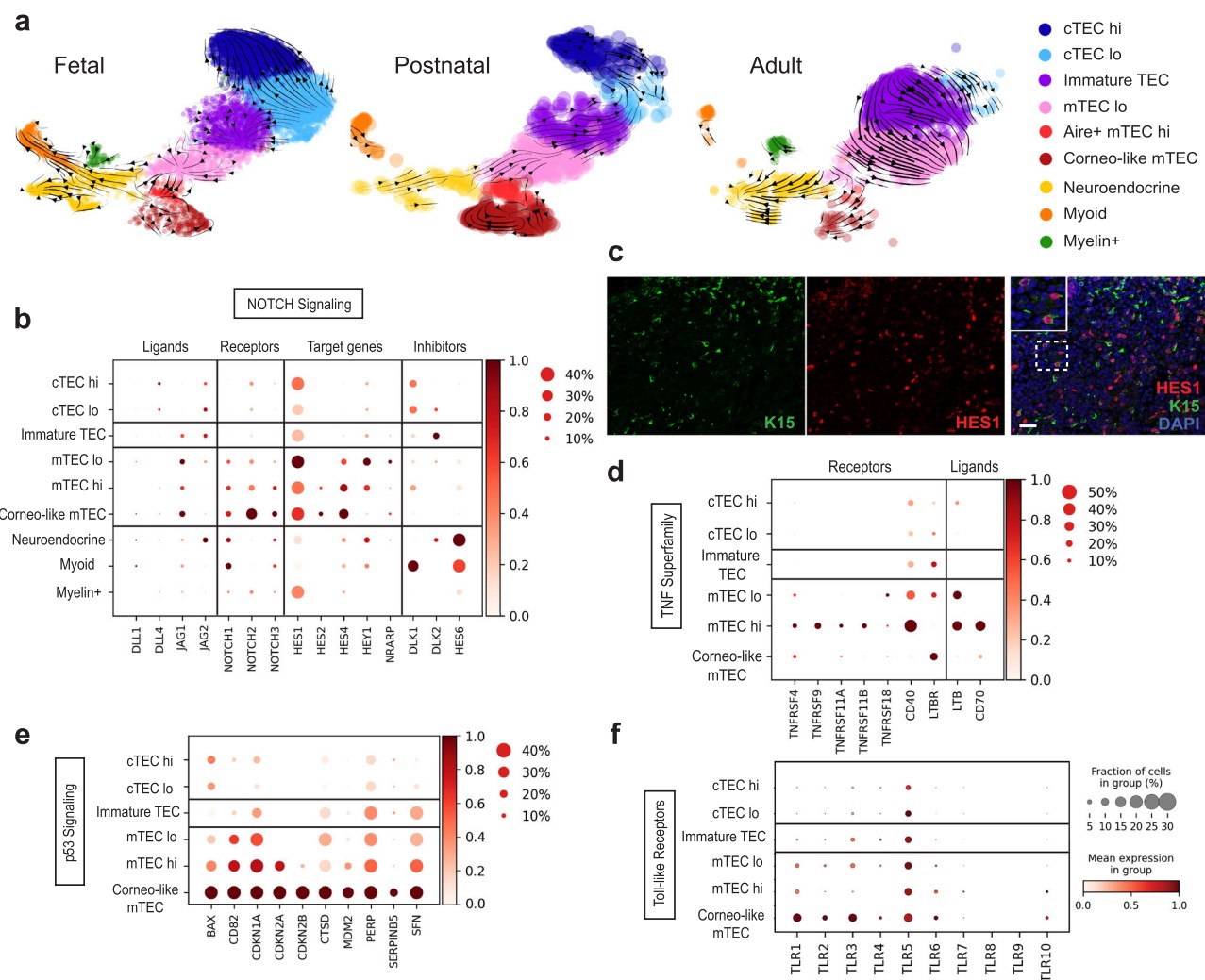

**Fig. 5 Lineage decisions within the thymic epithelial compartment. a** Velocity field projected on the UMAP plots of fetal, postnatal, and adult samples. **b** Dot plot depicting the relative level of expression of Notch signaling ligands, receptors, target genes, and inhibitors in epithelial subsets. **c** Immunofluorescence staining of human fetal thymus showing expression of HES1 (red) in medullary KRT15+ TECs (green). Scale bar, 100 μm. Staining was repeated twice with similar results. **d–f** Dot plots depicting the relative level of expression of selected TNF Superfamily (**d**), p53 (**e**), or Toll-like receptor signaling pathway (**f**) genes.

crossed to an Aire-GFP-reporter line (Aire-driven Igrp-GFP or Adig mouse model[43]) to facilitate quantification of Aire expression by flow cytometry. Mice were injected once with tamoxifen and thymi were harvested at 36 h and 5 weeks post-tamoxifen to analyze expression of the fluorescent reporters. As shown in Fig. 4h, 19% of TECs were labeled with the Ascl1-lineage tracing after 36 h, while the number dropped to 5% after 5 weeks. The percentage of Aire-expressing cells labeled with the Ascl1-lineage tracing also declined over time (Fig. 4h), implying that cells that maintain the pool of Aire+ mTEC$^{hi}$ are not expressing Ascl1. These data thus suggest that Ascl1+ mTEC$^{lo}$ and Ascl1+ mTEC$^{hi}$ are likely replenished from a pool of Ascl1-negative cells.

**Lineage decisions within the thymic epithelial compartment.** To better understand the relationship between the different epithelial subsets, we performed pseudotime analysis using RNA velocity[44]. This method, which considers both spliced and unspliced mRNA counts to estimate how mRNA levels evolve over time, can be used to predict potential directionality of transitions between cell states. Since murine TEC development differs significantly between embryonic and postnatal tissue, we analyzed

fetal, postnatal, and adult samples separately. Our analysis shows that cTECs$^{lo}$ give rise to cTECs$^{hi}$ in both fetal and postnatal tissues (Fig. 5a). The analysis also implied that cTECs$^{lo}$ can give rise to immature TECs in fetal samples while in the postnatal tissue, this relationship seems to be inverted with immature TEC transitioning back to cTEC$^{lo}$. As for mTECs, the analysis predicts that AIRE+ mTEC$^{hi}$ precedes both CCL21+ mTEC$^{lo}$ and corneocyte-like mTEC in fetal tissues while AIRE+ mTEC$^{hi}$ seem to give rise to corneocyte-like mTEC only in postnatal tissues.

Although it is clear that the epithelial compartment is comprised of many different subsets, the signals that control lineage specification still remain ambiguous. To identify pathways that might be regulating this lineage decision process, we analyzed genes differentially expressed between epithelial clusters (Supplementary Data 2). The Notch target gene *HES1* and Notch inhibitor *HES6* were identified as being enriched in mTEC$^{lo}$ and neuroendocrine/myoid clusters, respectively (Supplementary Data 2). Given the critical role of Notch signaling in regulating lineage choice in other tissues as well as its recently described role in TEC specification[45,46], we examined the expression of genes involved in this pathway, including ligands, receptors, target genes, and inhibitors. We found that *DLL4*, a key ligand promoting Notch1-dependent T cell

specification and maturation, and *JAG2* were the main ligands expressed in cTECs while *JAG1* was detected in immature TECs and mTECs (Fig. 5b and Supplementary Fig. 4a). Importantly, three of the receptors (*NOTCH1, NOTCH2, NOTCH3*) and many of their target genes (*HES1, HES2, HES4, HEY1,* and *NRARP*) were expressed at higher levels in mTECs, indicating that Notch signaling is more active in mTECs than cTECs. Immunofluorescence analysis demonstrated that HES1 protein is detected in KRT15+ TECs (Fig. 5c), confirming that Notch signaling is active in these TECs. The Notch inhibitor *DLK1* was also detected in cTECs while *DLK2* was found in immature TECs, implying that Notch signaling is reduced in these cells. Finally, expression of *HES6*, a negative regulator of *HES1*, was considerably higher in neuroendocrine and myoid cells, indicating that Notch activity is actively blocked in these epithelial subsets. Our analysis thus suggests that, in addition to its critical role in T cell specification, the regulation of Notch signaling likely affects cell fate outcomes in the epithelial compartment.

To further define TEC specification, we analyzed pathways that have been shown to regulate the development of CCL21+ mTEC^lo and AIRE+ mTEC^hi in mice. The maintenance of murine AIRE+ mTEC^hi cells is mediated by TNF receptor superfamily signals, including receptor activator for NF-κB (RANK) and CD40[47–50], as well as osteoprotegerin (OPG), which acts as a decoy receptor for RANKL[51]. In contrast, CCL21 + mTEC^lo depend on LTBR signaling[30]. Expression of a subset of these receptors was detected in CCL21+ mTEC^lo and corneocyte-like mTECs (*LTBR*), mTEC^hi (*TNFRSF11A* (also known as RANK), OPG (also known as *TNFRSF11B*), or mTEC^lo and mTEC^hi (*CD40*) (Fig. 5d and Supplementary Fig. 4b), confirming that similar pathways are likely controlling the differentiation of mTECs in humans. Intriguingly, other TNF receptors were also found in mTEC^hi (*TNFRSF4* (also known as OX40), *TNFRSF9* (also known as 4-1BB)) as well as some TNF ligands in mTEC^lo (*LTB*) or mTEC^hi (*LTB* and *CD70*) (Fig. 5d). Given the importance of LTBR signaling for the development of CCL21+ mTECs[30], FEZF2+ mTECs[52], and corneocyte-like mTECs[53], the observation that mTEC^lo as well as mTEC^hi express the ligand *LTB* suggest that the mTEC compartment might contribute to the differentiation of these TEC subsets, in addition to signals from thymocytes[53]. As for OX40 and 4-1BB, their function in mTECs is unknown while CD70 expression has been shown to promote regulatory T cell development in the murine thymus[54].

Finally, we identified genes from the p53 signaling pathway (*PERP, SFN, CTSD, CDKN2A, CDKN2B*) as well as many Toll-like receptors (*TLR1-6, TLR10*) that were upregulated in corneocyte-like mTECs (Fig. 5e–f, Supplementary Fig. 4c and Supplementary Data 2). In the murine thymus, p53 has been shown to be required for normal medullary TEC development[55]. Although it is not clear which subset of mTECs depends on this signaling pathway, our data identified corneocyte-like/post-AIRE mTECs, which have the highest levels of p53 activity, as an interesting candidate. As for Toll-like receptors, it is possible that this pathway regulates the differentiation of mTEC^hi cells into involucrin+ post-Aire cells, similar to what has been shown in mice[56]. Taken together, our analysis thus revealed important information on the process of cell fate commitment in the epithelial compartment.

**Characterization of rare medullary epithelial subsets.** To better understand the heterogeneity of the medullary compartment and gain insight into the relationship between mTECs and other epithelial subsets, we re-clustered mTECs, neuroendocrine, myoid, and myelin-expressing cells and increased the resolution to obtain eight clusters (Fig. 6a). Importantly, our analysis identified a population of ciliated cells (positive for *ATOH1, GFI1,*

*LHX3, FOXJ1*) and revealed that myelin+ cells closely resemble Schwann cells (*SOX10, MPZ, MBP, S100A1*) (Fig. 6a, b and Supplementary Data 4). Ciliated cells have been previously reported in the murine thymus[57,58] but their characterization has been limited to morphological observations. Our study thus provides the first transcriptomic data for these rare epithelial subsets. We also identified a cluster that contained cells with a signature characteristic of chemosensory tuft cells recently identified in the murine thymus[8,9] (*GNB3, TRPM5, GNAT3, PLCB2, OVOL3, POU2F3*) and cells with a signature reminiscent of the ionocyte population present in lung tissue[59,60] (*FOXI1, ASCL3, CFTR, CLCNKB*) (Fig. 6a, b and Supplementary Data 4). Since the presence of ionocytes in the thymus has not been reported previously, we confirmed that KRT8+/CFTR+ cells and TRPM2+/ CFTR+ were indeed found in the human thymic medulla of both fetal and postnatal tissue by immunofluorescence (Fig. 6c). Ionocytes were found as isolated cells in the medulla or as part of Hassall's corpuscles. In addition to ionocytes and thymic tuft cells[8], neuroendocrine cells and a subset of myoid cells were also detected in close proximity to Hassall's corpuscles (Fig. 6d). Although the presence of neuroendocrine and myoid cells had been previously reported, it is still unclear whether they originate from the TEC lineage or from neural crest cells that were incorporated in the tissue during embryogenesis. Intriguingly, we observed co-staining of the myoid marker desmin with cytokeratin+ cells in fetal tissue, suggesting that myoid cells might arise from epithelial cells during embryogenesis (Fig. 6e).

To validate our findings as well as to increase the number of rare cells in our analysis, we merged our medullary TEC subsets with their equivalent in the Park et al. dataset (mTEC I-IV, TEC (neuro), and TEC(myo)) (Supplementary Fig. 5a). The resulting clusters were annotated using markers described in Fig. 2b, c (Supplementary Fig. 5b). Tuft cells, ionocytes, and ciliated cells were found at all stages in both datasets, while myelin+ cells were only detected in our fetal and adult samples (Supplementary Fig. 5c, d). To better understand the relationship between tuft cells and ionocytes, we sub-clustered the population that contained both cell types and increased the resolution to separate the two subsets (Supplementary Fig. 5e, f). We next identified genes that were differentially expressed between tuft cells and ionocytes and compared this list to a set of markers characteristic of their counterpart in the human respiratory tract[61] (Supplementary Fig. 5g and Supplementary Data 5). While many genes were similar between thymic and pulmonary ionocytes (251 genes out of 500), the overlap between thymic tuft cells and pulmonary tuft-like cells was less pronounced (85 genes out of 500). Notably, many of the genes that were unique to human thymic tuft cells were also found in murine thymic tuft cells (*GNAT3, TRPM5, AVIL, GFI1B* as well as the taste receptor genes *TAS1R3, TAS2R4, TAS2R10,* and *TAS2R30*), suggesting that the cells are more similar to their murine counterpart than tuft-like cells found in the human respiratory tract.

Analysis of differentially expressed genes among medullary cells (Supplementary Data 4) identified the transcription factor *SOX2* as being highly expressed in ciliated cells. In addition, expression of *SOX2* was detected in corneocyte-like mTECs, neuroendocrine, and myelin+ cells (Fig. 6f). SOX2 has been shown to play a critical role in lineage specification, proliferation, and differentiation during embryogenesis and is also involved in maintenance of stem and progenitor cell populations in many adult tissues[62]. A role for this transcription factor in the postnatal thymus has, however, never been reported. Immunofluorescence analysis confirmed that SOX2 was expressed in Hassall's corpuscles as well as in a few isolated cells scattered in the medulla (Fig. 6g, h). SOX2 staining was observed in KRT8+ cells as well as in cells expressing KRT5 and/or KRT10, confirming

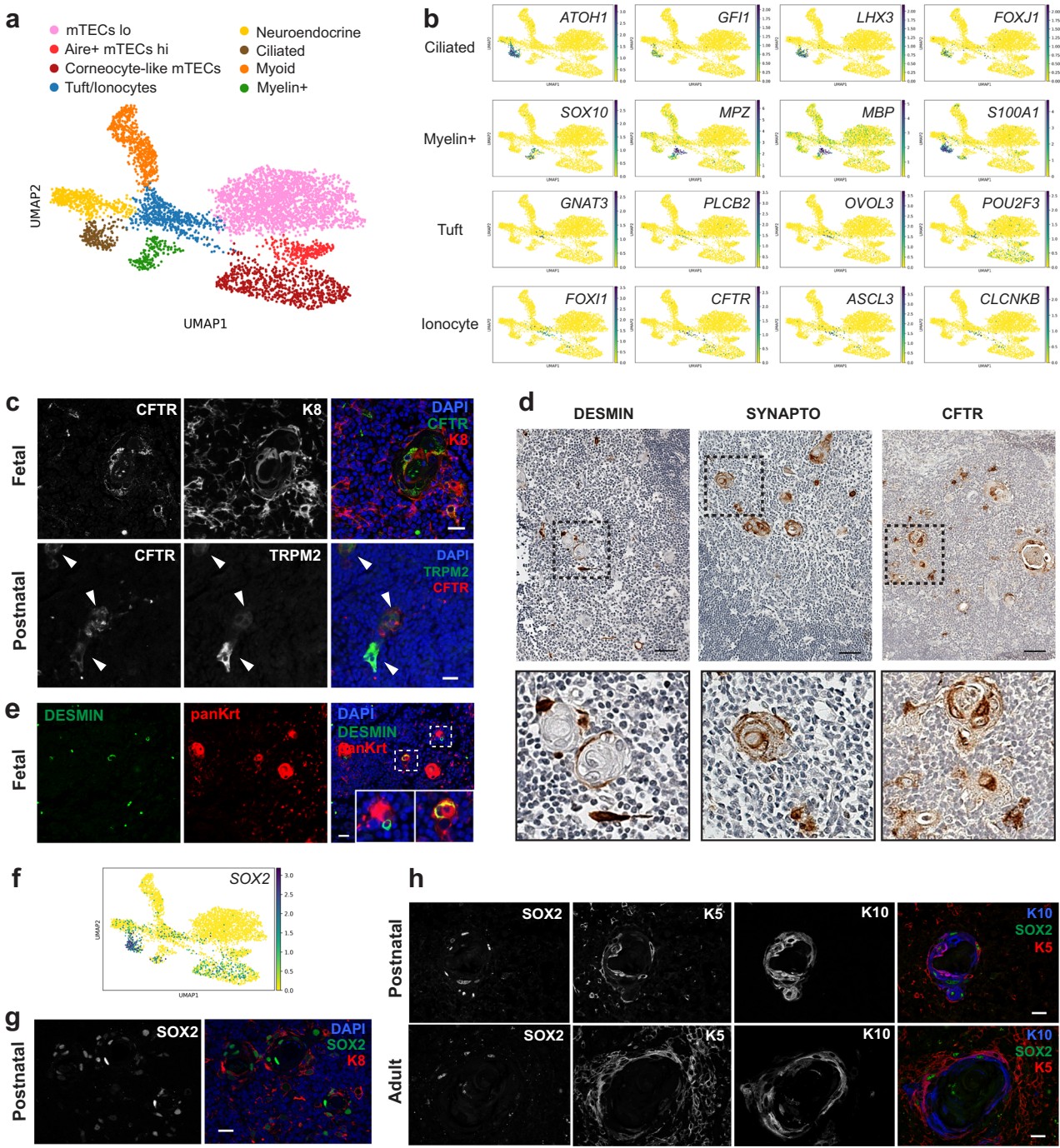

**Fig. 6 Tuft cells, ionocytes, ciliated cells, and myelin-expressing cells are present in the human thymic medulla. a** UMAP visualization of mTECs, neuroendocrine, myoid, and myelin-expressing cells sub-clustering. **b** UMAP visualization of the expression of marker genes used for cell cluster identification. **c** Immunofluorescence analysis of human fetal and postnatal thymus confirming the presence of ionocytes positive for KRT8 (red) and CFTR (green) or TRPM2 (green) and CFTR (red) in the medulla. Scale bars, 20 μm. **d** Immunohistochemistry staining for ionocytes (CFTR), neuroendocrine cells (SYNAPTO), and myoid cells (DESMIN). Scale bars, 100 μm. **e** Immunofluorescence analysis showing co-staining of desmin-expressing myoid cells (green) with a wide spectrum cytokeratin antibody (red) in human fetal thymus. Scale bar, 25 μm. **f** UMAP visualization of SOX2 expression in medullary epithelial cells. **g**, **h** Immunofluorescence staining for SOX2 in postnatal and adult thymus confirms expression of this transcription factor in the medulla. A subset of SOX2+ cells (green) co-expressed KRT8 (red) (**g**) or KRT5 (red) and/or KRT10 (blue) (**h**). Scale bars, 20 μm. Staining in **c**, **d**, **e**, **g**, **h** was repeated at least twice with similar results.

expression in different subsets of medullary epithelial cells (Fig. 6g, h). The observation that SOX2 is found in many cells, including most cells forming Hassall's corpuscles, is intriguing since this factor is often a marker of long-lived postnatal progenitor populations, suggesting these structures might be more dynamic than previously appreciated.

**Characterization of TSA expression**. Due to the heterogeneous nature of promiscuous gene expression in mTECs, analysis at the single-cell resolution provides an unprecedented opportunity to better understand how expression of TSAs is regulated in the human thymus. Studies in rodents have established that the autoimmune regulator Aire plays a crucial role in inducing

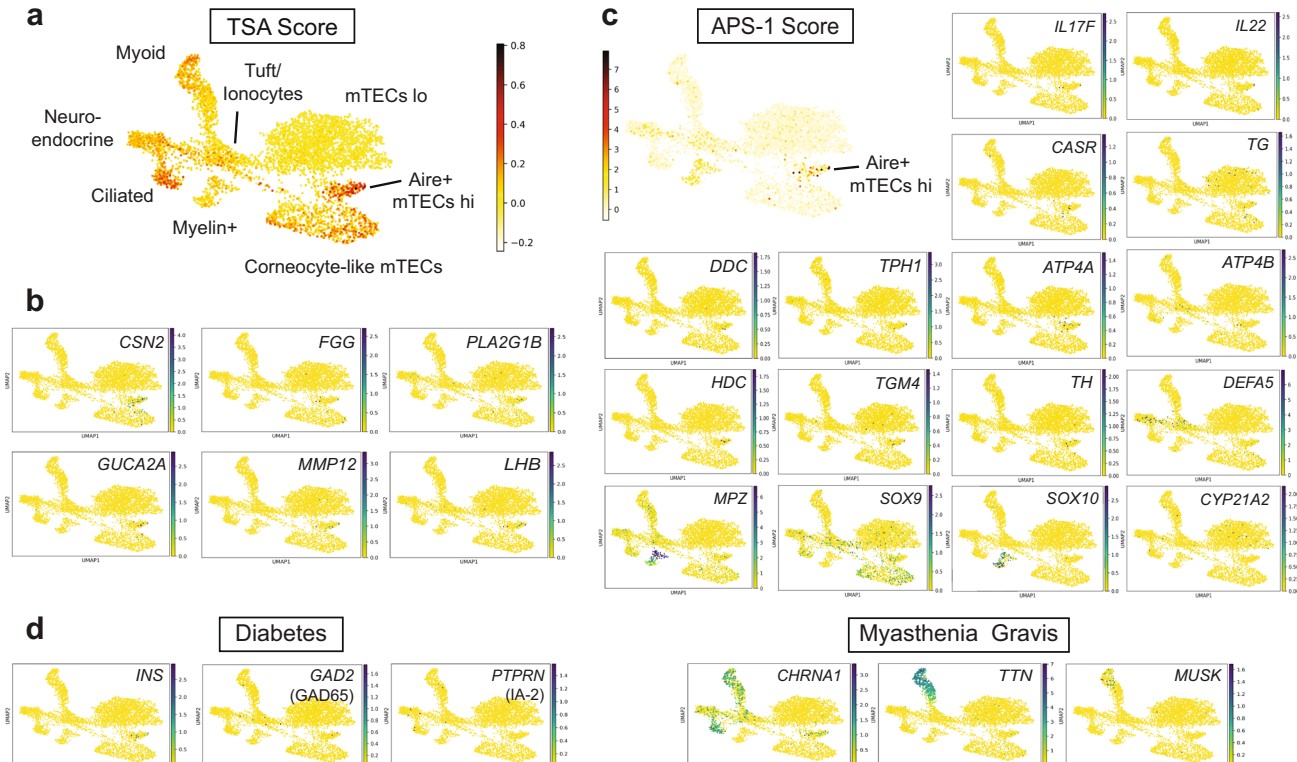

**Fig. 7 Characterization of tissue-specific antigen expression by human TECs. a** UMAP visualization of the average expression of tissue-specific antigens (TSA score) in medullary epithelial cells. **b** UMAP visualization of the expression of genes that positively correlate with a high TSA score in AIRE+ or corneocyte-like mTECs. **c** UMAP plots showing the expression of antigens eliciting autoantibodies in APS-1 patients. **d** Feature plots of antigens eliciting autoantibodies in type 1 diabetes and myasthenia gravis patients.

central tolerance by facilitating expression of thousands of TSAs in a subset of mTECs[63,64]. Although it is clear that loss-of-function mutations in the human AIRE gene result in a multi-organ autoimmune disease known as autoimmune poly-endocrinopathy candidiasis ectodermal dystrophy (APECED) or autoimmune polyglandular syndrome type 1 (APS-1)[65,66], direct evidence of TSA expression in human AIRE+ mTECs is lacking. To assess if AIRE+ mTECs expressed higher levels of TSAs than other epithelial subsets, we looked at the number of genes expressed per cell in each cluster (Supplementary Fig. 6a) and calculated a TSA score by averaging expression of a list of tissue-restricted genes (compiled by Sansom et al.[67]) and subtracted the average expression of a reference set of genes. The results were visualized using Uniform Manifold Approximation and Projection (UMAP) (Fig. 7a and Supplementary Fig. 6b). Importantly, this analysis confirmed that TSA expression was particularly enriched in human AIRE+ mTEC[hi], corneocyte-like mTECs, and ciliated cells when compared to other epithelial subsets. We also generated a list of the TSAs that were associated with a high TSA score in AIRE+ mTEC[hi] and corneocyte-like mTECs (Supplementary Data 6). The results include genes that have been previously associated with murine mTECs such as *CSN2, FGG,* and *PLA2G1B*, but also identified additional TSAs like *GUCA2A, MMP12,* and *LHB* (Fig. 7b and Supplementary Data 6). Next, we used a similar approach to analyze the expression of antigens known to elicit autoantibodies in APS-1 patients[68], which are predicted to be AIRE-dependent genes in humans. This analysis confirmed that APS-1 relevant genes were enriched in AIRE+ mTECs (Fig. 7c and Supplementary Fig. 6c). In addition, visualization of individual genes using UMAP revealed which antigens were mostly detected in AIRE+ and post-AIRE mTECs (*IL6, IL17F, IL22, CASR, TG, TPO, BPIFB1, DDC, TPH1, ATP4A,*

*ATP4B, HDC, TGM4, TH*). In contrast, other genes were mainly expressed in the neuroendocrine or myelin+ clusters (*DEFA5, SOX10, MPZ*) (Fig. 7c). This analysis thus provides additional evidence that AIRE-expressing cells play a critical role in the development of the APECED/APS-1 disease. To better understand the role of different epithelial subsets in the induction of immune tolerance in humans, we also analyzed expression of antigens that have been shown to play a role in organ-specific autoimmune diseases. As shown in Fig. 7d, antigens relevant to type 1 diabetes (T1D) like insulin (*INS*) and *GAD2* were found in rare AIRE+/post-AIRE cells while IA-2 (*PTPRN*) was detected in a few neuroendocrine cells. These data thus confirmed that, similar to the murine thymus, expression of insulin in the human thymus is likely AIRE-dependent. In contrast, genes coding for the acetylcholine receptor (*CHRNA1*), the muscle antigen titin (*TTN*), and *MUSK*, which are associated with the neuromuscular autoimmune disease myasthenia gravis, were predominantly found in the myoid, neuroendocrine, and ciliated subsets (Fig. 7d). While these cells likely do not directly present antigens to thymocytes due to low levels of HLA class I and II expression (Fig. 2d), it is possible that they participate in the induction of immune tolerance by providing peptides to antigen-presenting cells present in the thymic medulla. Our study thus provides an invaluable resource to define promiscuous gene expression in thymic stromal cells and will help shed light on how central tolerance is established in humans.

## Discussion

In this study, we generated a comprehensive single-cell database of human thymic stromal cells with a particular focus on the epithelial compartment. This work represents a major advance

beyond previous efforts, which focused mostly on mouse TECs or immune cell populations[9,12,31,69–71]. Our in-depth analysis of epithelial populations identified ionocytes as an additional subset of medullary epithelial cells and provided transcriptome information for rare subsets, including ciliated and Schwann cells that had only been described in histological analyses.

While it has been established that non-epithelial stromal cells are critical for TEC development and function, the specific contribution of each stromal cell type is not well understood. Our single-cell dataset provides invaluable information on factors expressed by the diverse thymic stromal populations and will enable a better understanding of the cross-talk between TECs and the rest of the stroma. For example, we identified a subset of postnatal mesenchymal cells that secrete many components of the WNT pathway, including the non-canonical WNT ligand *WNT5A* as well as molecules that can potentiate the WNT/Ca2+ pathway (*RSPO3* and *SFRP2*). Given that WNT signaling has been implicated as a critical regulator of Foxn1 expression[19], thymic cellularity[72,73], and migration of the thymus during development[74], our data points to this subset of mesenchymal cells as an important regulator of these processes. Notably, this population also expressed other critical regulators of epithelial proliferation and differentiation (*BMP4*, *IGF1*, *FGF7*, and *FGF10*)[21–23,75–77], further highlighting the importance of mesenchymal cells for TEC homeostasis. In addition, a recent study demonstrated that activin A signaling is critical for TEC maturation while follistatin, an inhibitor of this pathway, contributes to the accumulation of TEPCs by blocking differentiation of TECs[6]. It is intriguing that our analysis identified pericytes as the main source of activin A. Additional studies will be required to evaluate the role of this understudied population of stromal cells in regulating TEC differentiation. Notably, our analysis also revealed that myoid cells express high levels of follistatin, thus providing insight into why conditions such as myasthenia gravis, which affect myoid cell numbers, perturb human TEC differentiation.

A striking outcome of our study is the identification of CFTR+ ionocytes as an additional subset of epithelial cells found in the human thymic medulla. While ionocytes have been described in lung epithelium[59,60], their presence has not been previously reported in the thymus. Intriguingly, pulmonary ionocytes arise from basal cells who also give rise to neuroendocrine and tuft cells[59]. Given that these cell types are also present in the human thymus and that they were found in close proximity to each other in the medulla, with many subsets associated with Hassall's corpuscles, it raises the possibility that a similar progenitor exists in the thymus. This hypothesis is compatible with reports of thymomas comprising neuroendocrine differentiation[78] and thymic carcinomas containing tumor cells with a neuroendocrine phenotype[79,80]. It is also intriguing that myoid cells occur in different thymic tumors, including different types of histologic variants of thymoma and thymic carcinomas[81]. Tumors showing both rhabdomyoid and epithelial differentiation can also arise in the thymus[82], suggesting that there might be a common precursor that can give rise to both epithelial and myoid cells. Our transcriptome data showing a branching point between neuroendocrine and myoid cells as well as co-expression of epithelial and myoid markers by immunofluorescence in fetal tissue support this idea. Our work thus provides additional insight on the origin of epithelial subsets found in the human thymus.

Our analysis suggested a role for Notch signaling in the development of different thymic epithelial subsets. While Notch signaling has been extensively studied in the context of T cell commitment and epithelial differentiation in other tissues, a role for this pathway in TEC specification has only been recently reported[45,46]. In neuronal and muscle stem cells, high and sustained levels of Hes1 can inhibit cell differentiation by antagonizing master

regulators of cell fate like the proneural factor Ascl1 and regulator of myogenesis Myod1[83]. In contrast, when Hes1 expression oscillates, it activates stem cell proliferation by driving oscillations in Ascl1 and Myod1 expression through periodical repression cycles[83,84]. Alternatively, terminal differentiation is promoted when expression of Ascl1 or Myod1 is sustained while Hes1 expression and/or activity is inhibited. This mechanism is compatible with our observation that HES6, a HES1 inhibitor, is highly expressed in neuroendocrine and myoid cells. It is thus possible that HES6-mediated inhibition of HES1 allows stable expression of ASCL1 and MYOD1 in progenitor cells that will eventually differentiate into ASCL1+ neuroendocrine or MYOD1+ myoid cells. Although the mechanisms that lead to high Hes1 expression in progenitor cells are not clear, other signaling pathways like BMP have been shown to play an important role in regulating quiescence by upregulating Hes1 in neural stem cells[85] or promoting the degradation of ASCL1 in hippocampal stem cells[86]. Since there is evidence that BMP signaling promotes maintenance of thymic progenitors[6], it is possible that this pathway helps establish and maintain quiescence in TEPCs by upregulating HES1 to a level where it constantly suppresses expression of differentiation factors like ASCL1.

Finally, our study provides a valuable database of genes expressed in TECs that will allow a better understanding of promiscuous gene expression in the human thymus. For example, a significant subset of APS-1 antigens were present in AIRE+ mTECs, supporting the idea that these genes are AIRE-dependent in humans. Interestingly, some of the autoantigens that were not found in AIRE+ cells (*CYP21A2*, *CYP17A1*, *CYP11A1*) are often targeted by autoantibodies in thymoma patients but do not correlate with AIRE expression in thymoma samples, implying that they are not AIRE-dependent genes[87,88]. In addition to AIRE+ mTECs, other cell types most likely participate in induction of tolerance by providing antigens that can be presented by antigen-presenting cells like dendritic cells. For example, myoid cells likely participate in the induction of immune tolerance to muscle antigens. Indeed, many thymoma patients who typically lack myoid cells[89] develop myasthenia gravis (MG), an autoimmune disease of the neuromuscular junction characterized by auto-antibodies to the acetylcholine receptor (AChR) or other muscle antigens like titin (TTN)[90]. APS-I patients also typically don't have detectable autoantibodies against either AChR or TTN, suggesting that the expression of these antigens is not entirely AIRE-dependent[88]. Importantly, expression of AChR and TTN in our dataset was much higher in myoid cells compared to AIRE+ mTECs, supporting the idea that myoid cells are the main source of muscle antigens in the human thymic medulla. Our analysis thus provides additional information on the regulation of disease-relevant TSAs in the human thymus.

In summary, we created reference transcriptome maps for individual stromal cell types across multiple stages of life to better understand how the thymic microenvironment is established and maintained in aging. In addition to advancing our knowledge of human thymic development, this study provides evidence of greater heterogeneity among medullary TECs than was previously appreciated. This work also offers a platform to study the expression of disease-relevant antigens in different thymic subsets, thus providing insight on the relevance of this heterogeneity to the induction of immune tolerance and human autoimmune diseases.

## Methods

**Thymic tissue acquisition**. Human fetal thymic tissues were obtained from 19 to 23 gestational-week specimens under the guidelines of the Committee on Human Research (UCSF IRB)-approved protocols from the Department of Obstetrics, Gynecology and Reproductive Science, San Francisco General Hospital. Samples were obtained after legal, elective termination of pregnancy with written informed

consent for fetal tissue donation to biomedical research. Consent for tissue donation was obtained by clinical staff after the decision to pursue termination was reached by patients. Personal Health Information and Medical Record Identifiers/access is at no point available to researchers, and no such information is associated with tissue samples at any point. Pediatric tissues were obtained from patients undergoing corrective cardiothoracic surgery in accordance with protocols approved by the UCSF Human Research Protection Program Institutional Review Board (IRB #17-22928). Written informed consent was obtained from the patient's parents, guardians, or Legally Authorized Representatives before sample collection. Human adult thymic tissues were acquired from research-consented deceased organ donors at the time of organ acquisition for clinical transplantation through an IRB-approved research protocol with Donor Network West, the organ procurement organization for Northern California. All donors were free of chronic disease and cancer and were negative for hepatitis B/C and HIV. Tissues were collected after the clinical procurement process was completed, and stored and transported in University of Wisconsin (UW) preservation media on ice, and delivered at the same time as organs for transplantation. The study does not qualify as human subjects research, as defined by the UCSF IRB, as tissue samples were obtained from de-identified deceased individuals without associated personal health information (PHI). Demographic and clinical data were extracted from de-identified materials provided by Donor Network West.

**Mice.** Rosa26$^{CAG-stopflox-tdTomato}$ and Ascl1-cre$^{ERT2}$ (ref. [91]) mice were obtained from The Jackson Laboratory (RRID:IMSR_JAX:007914 and RRID:IMSR_JAX:012882). ADIG mice have been described previously[43]. Mice were maintained in the University of California San Francisco (UCSF) specific pathogen-free animal facility in accordance with the guidelines established by the Institutional Animal Care and Use Committee (IACUC) and Laboratory Animal Resource Center. Mice were maintained at a constant humidity between 30 and 70% and temperature 68 and 79 °F, under a 12-h light/dark cycle and had free access to food and water. All experimental procedures were approved by the Laboratory Animal Resource Center at UCSF. Mice aged 12–15 weeks were used for the lineage tracing experiments regardless of their sex. Tamoxifen (Sigma-Aldrich) was dissolved in corn oil (Sigma-Aldrich) and one 4 mg dose was administered by oral gavage with flexible plastic feeding tubes (Instech). All mice were euthanized by $CO_2$ inhalation followed by cervical dislocation according to IACUC guidelines.

**Tissue preparation.** Thymic tissues placed in RPMI (ThermoFisher) containing 100 μg/ml DNase I (Roche) were cut into small pieces using scissors. Tissue pieces were transferred into a gentleMACS C tube (Miltenyi) containing 10 ml of RPMI with DNAse. The gentleMACS Program m_spleen_02 was run three times. Thymic fragments were separated from the thymocytes-rich supernatant by centrifugation. Remaining fragments were transferred back to C tube with fresh RPMI with DNAse before running program m_spleen_01. The supernatant was removed and replaced with 10 ml of digestion medium containing 100 μg/ml DNase I and 100 μg/ml Liberase TM (Sigma-Aldrich) in RPMI. Tubes were moved to a 37 °C water bath and fragments were triturated every 5 min to mechanically aid digestion. At 30 min, tubes were spun briefly to pellet undigested fragments and the supernatant was discarded. Fresh digestion medium or accumax (Stemcell Technologies) was added to remaining fragments and the digestion was repeated for another 15–30 min until most pieces were digested. Supernatant from this second round of digestion was transferred to a tube containing cold MACS buffer (0.5% BSA, 2 mM EDTA in PBS) to stop the enzymatic digestion. If necessary, a third round of enzymatic digestion was performed on remaining fragments using accumax for an additional 5–10 min. Cells were pooled with the supernatant from the previous round of digestion and were passed through a 40-μm filter (Falcon). Some samples were treated with 2 ml of ACK lysing buffer (Lonza) for 5 min prior to stromal enrichment.

**Enrichment of stromal cells.** For fetal and postnatal tissues, single cells from digested tissue were resuspended in MACS buffer containing 10 μM of the ROCK inhibitor Y-27632 (Tocris). Human CD45 MicroBeads (Miltenyi) were used to deplete immune cells according to the manufacturer's instructions with the following modification: 5 μL of CD45 MicroBeads per $10^7$ total cells were added instead of 20 μL. LD columns were used for depletion and the CD45-negative fraction was collected in MACS buffer. Stromal cells from adult thymus were enriched using fluorescence-activated cell sorting (FACS). Blocking was done with human Fc receptor binding inhibitor monoclonal antibody (eBioscience) followed by staining for 20 min using human-specific antibodies against EPCAM (Clone 9C4, BioLegend) and CD45 (Clone HI30, Biolegend). After staining, cells were washed and resuspended in FACS buffer containing DAPI. Cells were sorted on BD FACS Aria II. Pre-gating was first done for live cells based the DAPI stain. The percentages of EPCAM$^+$, EPCAM$^-$CD45$^-$, and CD45$^+$ cells in each sample was determined by flow cytometric analysis or by calculating the ratio of cells in epithelial clusters, immune cluster, or other clusters over the total number of cells in the single-cell RNA-seq dataset.

**Single-cell RNA-seq and computational analysis.** Single cells were captured using the 10X Chromium microfluidics system (10X Genomics). The cells were encapsulated and barcoded cDNA libraries were prepared using the single-cell 3′ mRNA kit (v2 or v3; 10X Genomics). Single-cell libraries were sequenced using a NovaSeq 6000 (Illumina). The Cell Ranger software pipeline (10X Genomics, version 2.0.0, 2.1.1, or 3.0.2) was used to demultiplex cellular barcodes, map reads to the human genome (GRCh38) and transcriptome using the STAR aligner, and produce a matrix of gene counts versus cells. Single-cell data analysis was performed using Scanpy (version 1.4.4 or 1.6.0)[92]. Cell by gene count matrices of all samples were concatenated to a single matrix. Cells with 200–5000 detected genes and expressing <10% mitochondrial genes as well as genes expressed in ≥3 cells were retained for a total of 68,008 cells (23,328 cells from hFT 19.0 weeks, 17,984 from hFT 23.0 weeks, 1191 from hPT 6 days, 5865 from hPT 10 months, and 19,640 from hAT 25 yo) with an average of 4769 counts per cell. Counts were log transformed and total counts per cell were normalized. The dataset was filtered for highly variable genes (minimum mean = 0.0125, maximum mean = 3, and dispersion, 0.5 per gene) and variation caused by mitochondrial gene expression and cell cycle-dependent changes in gene expression were regressed out. BBKNN[13] was applied to correct donor-specific effects. K-nearest neighbor graphs were constructed (n_neighbors = 15) and clustering was performed using the Leiden algorithm with a resolution of 0.5. Clustering results was visualized using UMAP. Sub-clustering was performed by isolating clusters of interest with Scanpy and using the Leiden algorithm with a resolution of 0.7 for sub-clustering all TECs and 0.5 for sub-clustering mTECs. Characteristic gene signatures were identified by testing for differential expression of a subgroup against all other cells using a t-test with overestimated variance implemented in the tl.rank_genes_groups function of Scanpy. The TSA and APS-1 scores were calculated using the scanpy.tl.score_genes function. This function calculates the average expression of a set of genes subtracted with the average expression of a randomly selected reference set of genes[93]. TSA genes were identified using data from the GNF Mouse GeneAtlas as reported by Sansom et al.[67]. APS-1 genes were selected based on their association with autoantibodies in APS-1 patients[68]. The average expression level of scored genes was visualized using UMAP previously generated using Scanpy. To quantify the association between specific TSAs and the TSA score for AIRE+ and corneocyte-like mTECs, a Spearman correlation coefficient with associated p value was calculated using the spearmanr function from the SciPy statistical package. The list was filtered to keep genes expressed in at least five cells that had positive correlation coefficients >0.15 and p values < 0.25. Supplementary Data tables were prepared using Microsoft Excel.

**Comparison to published dataset.** The raw count matrix and annotated matrix for epithelial cells from the Park et al. dataset were downloaded from the Zenodo repository (https://doi.org/10.5281/zenodo.3711134). The dataset was processed through the same pipeline and combined with our epithelial dataset using Scanpy. Batch alignment was performed using the BBKNN algorithm. Clustering was performed using the Leiden algorithm with a resolution of 0.51 and results were visualized using UMAP. Sub-clustering was performed by isolating clusters of interest with Scanpy and using the Leiden algorithm with the resolution set at 0.25 for mTECs and 0.1 for ionocytes/tuft cells. Characteristic gene signatures were identified by testing for differential expression of tuft cells against ionocytes using a t-test with overestimated variance implemented in the tl.rank_genes_groups function of Scanpy.

**Trajectory analysis.** Trajectory analysis was performed using RNA Velocity[44]. Spliced and unspliced expression matrices were generated using the standard velocyto pipeline. The following steps were performed using the scVelo package[94]. The matrices were size-normalized to the median of total mRNA molecules across all cells. Genes were selected based on a threshold of a minimum of 20 expressed counts for both spliced and unspliced mRNA. The top 2000 highly variable genes were kept for further downstream analysis. Nearest neighbor graphs were calculated with 30 neighbors based on the normalized gene expression matrices from the original analysis. Velocity estimations were calculated using the standard scVelo pipeline and the resulting velocity graphs were projected onto the UMAPs previously generated using Scanpy.

**Immunofluorescence staining and imaging.** For immunofluorescence, tissues were fixed in 4% paraformaldehyde, washed with PBS, followed by overnight incubation with 30% (w/v) sucrose (Sigma-Aldrich) in PBS. Tissues were embedded in optimal cutting temperature compound (Tissue-Tek) and stored at −80 °C before sectioning on a cryostat (Leica). Slides were briefly rehydrated in PBS before permeabilization and blocking in CAS block (ThermoFisher) with 0.2% Triton X-100 (Sigma-Aldrich) followed by primary antibody staining at 4 °C overnight. When necessary, secondary antibody staining was performed at room temperature for 1 h. Sections were washed with PBS-Tween 0.1% before mounting with Pro-Long Diamond Antifade Mountant (ThermoFisher). Images were acquired on an Apotome microscope (Zeiss). Antibody details are provided in Supplementary Table 1.

**Immunohistochemistry**. Tissue was fixed in 4% paraformaldehyde (Thermo-Fisher), washed with PBS, and embedded in paraffin. Antigen retrieval was performed on rehydrated tissue by boiling sections in antigen retrieval Citra Solution (Biogenex). Sections were blocked for 30 min at room temperature using CAS-Block (ThermoFisher) with 0.2% Triton X-100 (Sigma-Aldrich), followed by incubation with primary antibody overnight at 4 °C. Staining with biotinylated secondary antibody was performed for 1 h at room temperature. Slides were developed using an ABC kit (Vector labs) and DAB kit (Vector labs) and counterstained with hematoxylin. Antibody details are provided in Supplementary Table 1.

**Flow cytometry**. For lineage tracing experiments, single-cell suspensions were prepared as previously described[8]. Briefly, mouse thymi were isolated, cleaned of fat and transferred to DMEM (UCSF Cell Culture Facility) containing 2% FBS (Atlanta Biologics) on ice. Thymi were minced with a razor blade and up to four thymi were pooled into a single digestion. Tissue pieces were moved to 15-ml tubes and vortexed briefly in digestion medium (DMEM containing 2% FBS, 100 μg/ml DNase I, and 100 μg/ml liberase TM). Fragments were allowed to settle before removing the medium and replacing it with fresh digestion medium. Tubes were moved to a 37 °C water bath and fragments were digested for 36 min with trituration with a glass Pasteur pipette every 6 min. At 12 min, tubes were spun briefly to pellet undigested fragments and the supernatant was moved to 20 ml of 0.5% BSA (Sigma-Aldrich), 2 mM EDTA (TekNova) in PBS (MACS buffer) on ice to stop the enzymatic digestion. This was repeated twice for a total of three 12-min digestion cycles, or until there were no remaining tissue fragments. The single-cell suspension was then pelleted and washed once in MACS buffer. Density-gradient centrifugation using a three-layer Percoll gradient (GE Healthcare) with specific gravities of 1.115, 1.065, and 1.0 was used to enrich for stromal cells. Cells isolated from the Percoll-light fraction, between the 1.065 and 1.0 layers, were resuspended in MACS buffer and counted. Cells were then incubated with Live/Dead Fixable Blue Dead Cell Stain (ThermoFisher) in PBS for 20 min at 4 °C followed by blocking with anti-mouse CD16/CD32 (24G2) (UCSF Hybridoma Core Facility) and 5% normal rat serum for 20 min at 4 °C. Cells were then washed in FACS buffer and stained for surface markers for 30–45 min at 4 °C. Flow cytometry data were collected on a LSRII Flow Cytometer (BD Biosciences) housed within the UCSF Single Cell Analysis Center using the FACSDiva software (BD Biosciences), analyzed using FlowJo software (TreeStar Software), and plotted using GraphPad Prism. Antibody details are provided in Supplementary Table 1.

**Reporting summary**. Further information on research design is available in the Nature Research Reporting Summary linked to this article.

## Data availability

Data that support the findings of this study have been deposited in the Gene Expression Omnibus (GEO) database under accession number GSE147520. The Park et al. dataset was downloaded from the Zenodo repository [https://zenodo.org/record/3711134#.YAZwrZNKhYg]. Source data are provided with this paper.

## Code availability

Single-cell RNA-sequencing analysis were done using publicly available pipelines as described in the Methods.

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

## Acknowledgements

We would like to thank members of the Anderson and Hebrok laboratories for helpful discussions and M. Cheng for critical reading of the manuscript. We would like to

acknowledge the donors and their families who generously donated thymic tissue for this study as well as the expert technical assistance from the personnel of the Institute for Human Genetics Core. Imaging and flow cytometry experiments were supported by resources from the UCSF Diabetes and Endocrinology Research Center (DRC) and UCSF Flow Cytometry Core (NIH Diabetes Research Center grant P30 DK063720). J.L.B. was supported by a T32 FAVOR training grant (project ID: 1T32AI125222-01). C.N.M. was supported by a National Institute of General Medical Sciences (NIGMS) Molecular and Cellular Immunology Program award (grant #T32AI00733430). L.E.B. was supported by an Achievement Rewards for College Scientists (ARCS) Foundation Scholar Award. J.M. G. was supported by the UCSF Sandler Fellows PSSP grant and the ASTS-Astellas Fellowship in Transplantation grant, and adult thymic tissue acquisition was courtesy of an approved research grant with Donor Network West. T.D.B was supported by an award from the Burroughs Wellcome Fund Preterm Birth Initiative and NIH grant K08 HD067295. This work was supported by a UCSF Diabetes Center Innovation Award to A.V.P. and NIH grant U01 DK107383 (A.V.P. and M.S.A.).

## Author contributions

The study was designed by J.L.B., C.J.Y., M.S.A., and A.V.P. Tissue acquisition was coordinated by J.L.B., T.D.B., and J.M.G. Tissue preparation and sequencing was performed and supervised by J.L.B., L.E.B., J.B.S., and A.V.P. Analysis of scRNA-seq data was done by J.L.B., N.T.C., J.G., and A.V.P. and supervised by V.N. and C.J.Y. Lineage-tracing experiments were designed and performed by C.N.M. Immunofluorescence experiments were done by J.C., D.B., and A.V.P. The manuscript was written by A.V.P. and reviewed by J.L.B., C.N.M., J.M.G., C.J.Y., and M.S.A.

## Competing interests

The authors declare no competing interests.
