## [Peer Review File · Nature Communications]

REVIEWER COMMENTS

Reviewer #1 (Thymocyte development, transcription regulation) (Remarks to the Author):

Bautista et al perform a transcriptional analysis of single cells from the human thymus, from fetal to adult stages of life, with a focus on thymic epithelial cells (TEC). Their single cell approach revealed previously unreported TEC heterogeneity in the human thymus, such as ionocytes and ciliated mTEC subsets, whilst identifying novel markers that could be used to isolate some of these newly identified populations. The work supports and extends data recently published (Park et al, Science 2020) that also used single cell RNA sequencing of human TEC across different ages to describe cell populations in the human thymus. Some of the authors' conclusions are supported by immunohistochemistry on human thymus sections, and a new mouse model is characterized to establish that the immature TEC population described here does not possess long-term self-renewal ability.

The experiments and bioinformatic analysis are well executed. However, there are also concerns, detailed below:

A major concern with this manuscript is that only one adult sample is included in the analysis, with relatively few TEC. Claims about underrepresentation of different TEC populations in the adult thymus, or any suggested differences between the composition of fetal and adult samples, seem premature. Ideally authors would validate their findings by increasing their sample sizes, particularly at the adult stage. The authors could examine data published from the Park et al manuscript already referred to (Science 367, 868, 2020). If the alterations in populations between fetal and adult samples were also observed in this data set, this would strengthen the conclusions. Additionally, gene signatures of the newly identified adult TEC populations may also be detectable in both data sets, strengthening the conclusions of the present study.

Additional points:

The figures, although attractive, seem unnecessarily difficult to follow. Labels in Figure 1D should be included in Figure 1F, and elsewhere as possible. Labels in Figure 2B should be used on all subsequent figures wherever possible. The use of a color bar is not adequate, especially when colors assigned in one figure are used in subsequent figures without labels.

Additional violin plots of expression of genes of interest across different ages, for example in Figure 3C with KRT15 and ASCL1, would be informative.

A role for Notch signaling in TEC is recently described (PMID: 32467237, 32467240) and should be referenced.

The use of the *Ascl1* *Ert2*Cre mouse strengthens the manuscript. However, the authors only display expression in total TEC or Aire+ mTEC. They do not display expression in the cTEC population, nor look at earlier time points (such as fetal or neonatal stages) to confirm *Ascl1* expression differs in the cortex and medulla at different stages of development, as stated in the text. These results should be presented. If possible, looking post 5 weeks of Cre induction would be worthwhile, to determine whether the small % of *Ascl1*-labeled TEC persist long-term.

One of the most interesting and novel findings of this manuscript is the identification of the 'immature TEC' population in the human thymus, Figure 2 B. However, in Figure 5 A, when TEC heterogeneity is explored in greater depth at a higher resolution, this population is removed from the analysis. As it is unclear if this population is lineage specific (cTEC or mTEC) and if this population could potentially contain a progenitor TEC subset. Additional analysis of this population would strengthen the manuscript.

Validation of any of the new markers of TEC heterogeneity by flow cytometry would also strengthen the author's findings. For example, can antibody staining for flow cytometry be performed for ASCL1, or Krt15?

EpCAM vs CD45 profiles by flow cytometry should be displayed for samples at all ages, not just the

adult sample (Figure 1B), so the frequency of populations is clear to the reader.

Reviewer #2 (Thymocyte development, thymic epithelial cells) (Remarks to the Author):

Overall comments

This is an essential resource paper for the thymic research community presenting a relatively unbiased dataset that interrogates human thymic stromal cell populations isolated from embryonic, neonatal and young adult thymi. The manuscript presents some interesting insight into the subpopulations that comprise the thymic stroma, identifies based on its transcriptome a seemingly new thymic epithelial cell state. However, the data is built on a relative low number of cells analysed per age group, orthogonal evidence for some of the rarer populations identified by the authors is lacking (e.g. myelin-expressing cells,) and the true ability of some cell populations to act as antigen-presenting cells is incompletely investigated (e.g. myoid cells). A detailed analysis about the developmental interrelationship between distinct TEC subpopulations is unfortunately missing which renders this report significantly less informative. Moreover, the data presented should furthermore be directly compared to the classification and gene expression profiles of other human (e.g. Park et al) and mouse studies (Bornstein et al.) so that it is clear which populations reported here correspond to which populations previously reported. The work presented could also benefit from a more in-depth bio-computational analysis including the analysis of developmental trajectories to establish possible precursor progeny relationships, and their changes over time to better highlight the differences in the stromal composition with age. With these additions, the manuscript could form an important source of information within the field of thymus biology but in its present format it is mainly a description of transcriptionally defined stromal subpopulations (most of which have previously been described, though not necessarily in human tissue).

Specific Comments and Questions:

1. Line 49: The concept that only medullary thymic epithelial cells effect negative selection is incorrect and should be edited (see for example: <https://www.pnas.org/content/110/12/4679>).
2. Line 82: It would be instructive to detail which murine data set was analysed to make the statement that "Ionocytes" were not identified in mice.
3. Line 101: Could the authors please comment on how many cells are retained post-QC filtering and whether there are difference in the number of cells analysed for each age/time point?
4. Line 103 and Figure 1D: Though three different TEC clusters are identified, a description is lacking how they are defined and to which population of antibody defined TEC populations they belong.
5. Lines 103-107: In many cases only one marker appears to have been used to identify cellular identity. Have the authors used any other datasets with more comprehensive marker gene sets to validate these identities? The authors will also need to demonstrate that all of the markers chosen to identify the different subpopulations are not expressed in other populations, or where possible define expression levels using bulk sequence analyses of specific subpopulations.
6. Figure 1 A + B: The authors should state the efficiency in depleting CD45-positive cells by MACS and FACS and quantify their frequency as the CD45+ contamination is substantial and surprisingly contains CD45+ cells from all age groups.
7. Figure 1D should provide a stacked bar graph to display the frequency of the individual cell populations for the individual ages of thymic tissue analysed.
8. Figure 1 H: FN1 stain is shown, it is unclear which cell population is identified. The Figure legend should state which TEC are marked by K15 expression.
9. Lines 118-120: Ligands previously associated with TEC differentiation are visualised. The authors should demonstrate or at least discuss whether the reciprocal receptors are expressed on TEC and, if

so, whether these are expressed in particular subpopulations?

10. Lines 120-123: There does not appear to be a p-value quoted to support these data.

11. Line 147: The authors refer to a population of cTEC_{lo} cells. It is unclear whether these cells can also be identified by surface phenotyping and whether they refer to "low MHC class II cell surface expressing cTEC when analysed by FACS and whether there are differences in the frequency of these cells between thyme of different ages. The authors should provide further details.

12. Lines 157-159: Without any functional read-out of "immature TEC", how can the authors be sure that this group of cells is unable to compensate in some respects for the loss of classical, mature TEC? As suggested, have the authors independently verified (see line 179) whether CDH13 unequivocally identifies immature TEC that bear at bulk the gene expression profile suggested to be characteristic of these cells as a means to verify the transcriptionally defined TEC subpopulations and link them to flow cytometrically identified TEC subsets.

13. Line 182: The authors make an interesting observation: the transcription level of a few genes appear to be higher in adult TEC when compared to foetal and neonatal cells. It would be helpful to know whether increased levels of transcripts correlate with heightened protein expression and in which pathways these gene products are likely to be involved so as to explain their putative role in thymus involution and extend their observation beyond a mere correlation to time. As these observations concern the so-called immature TEC, it would again be instructive to know the nature of the cells' developmental trajectory.

14. Line 189 and Figure 3D: The statement that K15+ TEC are positioned at the cortico-medullary junction is not supported by the data shown in the figure and thus does not provide the spatial correlate observed in the mouse for precursor cells that give rise to mTEC.

15. Line 196 and following: The expression of ASCL1 is an interesting observation. It would be important for the authors to demonstrate that this gene product is not expressed as part of an AIRE-independent promiscuous gene expression programme in cTEC and mTEC and that known ASCL1 target genes are co-expressed in ASCL1+ TEC.

16. Line 217 - 221: Are the authors in a position to comment on the labelling of neuroendocrine cells, their frequency and the change in labelled cells over time.

17. Line 246: Could the authors explain how they selected which genes from each pathway to display? Are there any measures of significance that can be attached to particular pathways in subtypes of TEC?

18. Line 269: The paragraph's conclusion is an overstatement of the data shown and known in the literature and should thus be edited to better represent what is shown as novel data.

19. Line 276: Details should be provided on what basis and by which computational means the re-clustering was achieved.

20. Lines 278-280: These novel epithelial populations are interesting. How certain are the authors that these cells are genuinely epithelial in nature and in the case of ionocytes indeed separate from Tuft cells and not just a mere difference in cell state? An immunofluorescent validation of these cells within thymic sections would be further helpful to characterize these cells. This would seem to be a sensible approach to validating these new populations. Could the authors comment on why this population drops out completely in the post-natal sample?

21. Line 286: It would be informative to show the differential gene expression between thymic ionocytes and Tuft cells and between these populations and their respective counterparts situated in the respiratory tract and the GI tract. This information would also provide further verification regarding the nature of these cells.

22. Lines 324-326: Given the relatively shallow sequencing depth here, could the authors comment on how many TSA are detected overall and in each individual TEC subpopulation? How many AIRE-induced genes are driving the TSA score?

23. Line 342: This is interesting data, although I am surprised that much can be seen with such shallowing sequencing depth. Could the authors please include cTEC to demonstrate a lack of TSA expression in these cells too? How are the authors identifying some of the more infrequently expressed TSA as being specific to individual TEC subtypes? Are the reported TSA enrichments

associated with a significant differential expression?

24. Lines 348-351: How sure are the authors that the myoid cells are not utilising these genes for other cellular functions and are truly capable of acting as antigen-presenting cells? This has been a controversy in the literature and could potentially be at least partially answered by close interrogation of the single cell RNA-seq dataset presented here.

25. What is the median number of genes per cell detected overall and by cluster? This would be important in enabling critical interpretation of the deeper phenotyping of TSA presented here.

26. Have the authors considered using RNA-based lineage tracing analyses, such as RNA Velocity, to identify differentiation trajectories within their data as suggested in several remarks made above?

Reviewer #3 (Transcriptome analyses, systems immunology) (Remarks to the Author):

The authors of this manuscripts have employed a single cell transcriptomics approach for the characterization of human thymic epithelial cell populations. They examined using this unbiased approach the composition of tissues from subjects corresponding to different developmental/age groups. Validation experiments were also conducted as follow on to some of the most original findings.

The manuscript is well written and well presented, which makes the description of the work and findings easy to follow. The findings follow in the footsteps of earlier studies employing scRNAseq to investigate Thymic T-cell development, but those either addressed primarily T-cell development or focused on embryonic tissues. Most of those studies are discussed in this work. One notable exception being the recent paper of Voboril et al in Nat. Commun. (PMID: 32398640). It is not directly "competing" with their effort but it might still be worth discussing. The authors should also be commended for making their data public via the NCBI Gene Expression Omnibus.

The only significant weakness stems from the fact that data and conclusions presented are based on profiles generated from at most two subjects per age group, and in the case on adults from only one subject. The fact that they examine something as fundamental as cell populations, without the need to quantify changes in relative abundance of these populations means that doing so may be adequate. However it remains somewhat unsettling to be drawing such general conclusions based on a single individual. This limitations should at least be acknowledged and discussed.

It is otherwise excellent work. Here are a couple of additional minor comments.

- In the abstract, Line 36, the authors refer to "TEC" without giving a definition for the abbreviation.

- Line 185: "the expression of KRT15 in the thymus has not been reported before " It is mentioned among a number differentially expressed in Thymic Epithelial Progenitors (although its significance was not highlighted investigated) :

<https://www.sciencedirect.com/science/article/pii/S2213671114001167>

- Generally, the authors have not discussed any limitations of the study (and specifically regarding the point mentioned above).

Manuscript : **NCOMMS-20-20031A**

Response to reviewers:

Reviewer #1 (Thymocyte development, transcription regulation) (Remarks to the Author):

Bautista et al perform a transcriptional analysis of single cells from the human thymus, from fetal to adult stages of life, with a focus on thymic epithelial cells (TEC) Their single cell approach revealed previously unreported TEC heterogeneity in the human thymus, such as ionocytes and ciliated mTEC subsets, whilst identifying novel markers that could be used to isolate some of these newly identified populations. The work supports and extends data recently published (Park et al, Science 2020) that also used single cell RNA sequencing of human TEC across different ages to describe cell populations in the human thymus. Some of the authors' conclusions are supported by immunohistochemistry on human thymus sections, and a new mouse model is characterized to establish that the immature TEC population described here does not possess long-term self-renewal ability.

The experiments and bioinformatic analysis are well executed. However, there are also concerns, detailed below:

A major concern with this manuscript is that only one adult sample is included in the analysis, with relatively few TEC. Claims about underrepresentation of different TEC populations in the adult thymus, or any suggested differences between the composition of fetal and adult samples, seem premature. Ideally authors would validate their findings by increasing their sample sizes, particularly at the adult stage. The authors could examine data published from the Park et al manuscript already referred to (Science 367, 868, 2020). If the alterations in populations between fetal and adult samples were also observed in this data set, this would strengthen the conclusions. Additionally, gene signatures of the newly identified adult TEC populations may also be detectable in both data sets, strengthening the conclusions of the present study.

We thank the reviewer for the positive comments. As suggested, we used the data from the Park et al. manuscript to increase our sample size and validate our findings. The results are presented in Supplementary Figures 2-5. We verified that functional TECs were also reduced in their adult samples (Supplementary Figure 2g) and confirmed increased expression of a subset of immature TEC genes over time (Supplementary Figure 3a). The presence of tuft cells, ionocytes, and ciliated cells was also validated in their dataset (Supplementary Figure 5c-d), thus supporting the conclusions from our study.

Additional points:

The figures, although attractive, seem unnecessarily difficult to follow. Labels in Figure 1D should be included in Figure 1F, and elsewhere as possible.

Labels have been added to Figure 1f.

Labels in Figure 2B should be used on all subsequent figures wherever possible. The use of a color bar is not adequate, especially when colors assigned in one figure are used in subsequent figures without labels.

Labels have been added to Figures 2d, 2e, and 3f (previously Figure 3a).

Additional violin plots of expression of genes of interest across different ages, for example in Figure 3C with KRT15 and ASCL1, would be informative.

We agree with the reviewer and added violin plots of KRT15 and ASCL1 expression across different ages to Figure 3 (3g for KRT15 and 3j for ASCL1).

A role for Notch signaling in TEC is recently described (PMID: 32467237, 32467240) and should be referenced.

The references have been added to the text.

The use of the Ascl1 Ert2Cre mouse strengthens the manuscript. However, the authors only display expression in total TEC or Aire+ mTEC. They do not display expression in the cTEC population, nor look at earlier time points (such as fetal or neonatal stages) to confirm Ascl1 expression differs in the cortex and medulla at different stages of development, as stated in the text. These results should be presented.

While we agree with the reviewer that showing expression of Ascl1 in cTECs would be informative, our flow cytometry panel did not include markers to differentiate between cTECs and mTECs. To address this point, we instead stained for endogenous Ascl1 expression in WT mice, in combination with Krt5 as a marker to delineate the medulla. As shown in Supplementary Figure 3c, we found that, in contrast to what we observed in the human thymus, Ascl1 was not found in the cortex of neonatal murine thymus.

If possible, looking post 5 weeks of Cre induction would be worthwhile, to determine whether the small % of Ascl1-labeled TEC persist long-term.

We unfortunately did not have mice available to perform this long-term experiment.

One of the most interesting and novel findings of this manuscript is the identification of the 'immature TEC' population in the human thymus, Figure 2 B. However, in Figure 5 A, when TEC heterogeneity is explored in greater depth at a higher resolution, this population is removed from the analysis. As it is unclear if this population is lineage specific (cTEC or mTEC) and if this population could potentially contain a progenitor TEC subset. Additional analysis of this population would strengthen the manuscript.

We agree with the reviewer's suggestions and have analyzed the immature cells at a higher resolution to gain more insights on the nature of these cells. The results are presented in Figure 3a-d and Supplementary Table 3). We also performed RNA Velocity analysis to help clarify the lineage relationship between these immature TECs and cTEC/mTEC populations. The results are included in Figure 4a.

Validation of any of the new markers of TEC heterogeneity by flow cytometry would also strengthen the author's findings. For example, can antibody staining for flow cytometry be performed for ASCL1, or Krt15?

As suggested by the reviewer, we validated that TECs expressed KRT15 by flow cytometry. The results have been added to Figure 3i.

EpCAM vs CD45 profiles by flow cytometry should be displayed for samples at all ages, not just the adult sample (Figure 1B), so the frequency of populations is clear to the reader.

A stacked bar graph showing the frequency of Epcam+, CD45+ and Epcam-CD45- for each sample has been added to Figure 1b and additional flow plots are shown in Supplementary Figure 1a.

Reviewer #2 (Thymocyte development, thymic epithelial cells) (Remarks to the Author):

Overall comments

This is an essential resource paper for the thymic research community presenting a relatively unbiased dataset that interrogates human thymic stromal cell populations isolated from embryonic, neonatal and young adult thymi. The manuscript presents some interesting insight into the subpopulations that comprise the thymic stroma, identifies based on its transcriptome a seemingly new thymic epithelial cell state. However, the data is built on a relative low number of cells analysed per age group, orthogonal evidence for some of the rarer populations identified by the authors is lacking (e.g. myelin-expressing cells,) and the true ability of some cell populations to act as antigen-presenting cells is incompletely investigated (e.g. myoid cells). A detailed analysis about the developmental interrelationship between distinct TEC subpopulations is unfortunately missing which renders this report significantly less informative. Moreover, the data presented should furthermore be directly compared to the classification and gene expression profiles of other human (e.g. Park et al) and mouse studies (Bornstein et al.) so that it is clear which populations reported here correspond to

which populations previously reported. The work presented could also benefit from a more in-depth bio-computational analysis including the analysis of developmental trajectories to establish possible precursor progeny relationships, and their changes over time to better highlight the differences in the stromal composition with age. With these additions, the manuscript could form an important source of information within the field of thymus biology but in its present format it is mainly a description of transcriptionally defined stromal subpopulations (most of which have previously been described, though not necessarily in human tissue).

We thank the reviewer for recognizing the value of our manuscript and have made substantial edits to address the concerns raised. The details are described below.

Specific Comments and Questions:

1. Line 49: The concept that only medullary thymic epithelial cells effect negative selection is incorrect and should be edited (see for example: <https://www.pnas.org/content/110/12/4679>).

We agree with the reviewer and have edited the text from 'mTECs are required for the deletion of autoreactive cells' to 'mTECs participate in the deletion of autoreactive cells' to clarify this point.

2. Line 82: It would be instructive to detail which murine data set was analysed to make the statement that "Ionocytes" were not identified in mice.

The source of the data (Bornstein et al. and Kernfeld et al.) has been added to the text.

3. Line 101: Could the authors please comment on how many cells are retained post-QC filtering and whether there are difference in the number of cells analysed for each age/time point?

The total number of cell post-QC filtering is included in the Results and Methods (68008 cells) and the specific numbers for each age have been added to the 'Single-cell RNA-seq and computational analysis' section of the Methods (23328 cells from hFT 19.0 wks, 17984 from hFT 23.0 wks, 1191 from hPT 6 days, 5865 from hPT 10 months, and 19640 from hAT 25 yo).

4. Line 103 and Figure 1D: Though three different TEC clusters are identified, a description is lacking how they are defined and to which population of antibody defined TEC populations they belong.

We added UMAP plots of cTECs and mTECs markers used to identify TECs with antibodies (FOXP1, PSMB11/ β 5t, LY75/CD205, CLDN4, AIRE, IVL) as well as other epithelial cell markers (NEUROD1 and MYOD1 to identify neuroendocrine and myoid cells, respectively) to Supplementary Figure 1d. These new data should help readers identify where specific TEC subpopulations fall in the UMAP of all stromal cells.

5. Lines 103-107: In many cases only one marker appears to have been used to identify cellular identity. Have the authors used any other datasets with more comprehensive marker gene sets to validate these identities? The authors will also need to demonstrate that all of the markers chosen to identify the different subpopulations are not expressed in other populations, or where possible define expression levels using bulk sequence analyses of specific subpopulations.

Cluster cell identity was assigned by manual annotation using multiple known marker genes as well as computed differentially expressed genes (DEGs). In addition to the list of DEGs provided in Supplementary table 1, we now provide at least three known markers for each cluster in the text, as well as a new matrix plot in Supplementary Figure 1b showing specific expression of these markers in the indicated subpopulations.

6. Figure 1 A + B: The authors should state the efficiency in depleting CD45-positive cells by MACS and FACS and quantify their frequency as the CD45+ contamination is substantial and surprisingly contains CD45+ cells from all age groups.

We added a graph with the frequency of Epcam+, CD45+ and Epcam-CD45- in each sample (Figure 1b). These data show that while there was a subset of CD45+ cells in all samples, the majority of the cells analyzed were CD45-.

7. Figure 1D should provide a stacked bar graph to display the frequency of the individual cell populations for the individual ages of thymic tissue analysed.
As suggested, a stacked bar graph has been added to Supplementary Figure 1c.
8. Figure 1 H: FN1 stain is shown, it is unclear which cell population is identified. The Figure legend should state which TEC are marked by K15 expression.
We added that K15 marks immature TECs and mTECs to the legend.
9. Lines 118-120: Ligands previously associated with TEC differentiation are visualised. The authors should demonstrate or at least discuss whether the reciprocal receptors are expressed on TEC and, if so, whether these are expressed in particular subpopulations?
Matrix plots showing the expression of different receptors in stromal subsets and TEC subsets has been included in Supplementary Figure 1e and 1f.
10. Lines 120-123: There does not appear to be a p-value quoted to support these data.
The p values from the differentially expressed genes analysis are provided in Supplementary Table 1.
11. Line 147: The authors refer to a population of cTEC_{lo} cells. It is unclear whether these cells can also be identified by surface phenotyping and whether they refer to “low MHC class II cell surface expressing cTEC when analysed by FACS and whether there are differences in the frequency of these cells between thyme of different ages. The authors should provide further details.
As mentioned in the text, cTEC_{lo} refers to low levels of functional genes, including MHC class II. A stacked bar graph showing the frequencies of the different cell types across ages has been added to Supplementary Figure 2a.
12. Lines 157-159: Without any functional read-out of “immature TEC”, how can the authors be sure that this group of cells is unable to compensate in some respects for the loss of classical, mature TEC?
We do not state that immature TECs are unable to compensate in any way but rather acknowledge that they lack expression of genes critical for their function, including HLA class II, PSMB11, AIRE, and CCL21 thus likely impairing their ability to support normal thymopoiesis.
- As suggested, have the authors independently verified (see line 179) whether CDH13 unequivocally identifies immature TEC that bear at bulk the gene expression profile suggested to be characteristic of these cells as a means to verify the transcriptionally defined TEC subpopulations and link them to flow cytometrically identified TEC subsets.
While it would have been interesting to verify the expression of CDH13 by flow cytometry, it has proven difficult to get access to fresh tissue during the COVID pandemic. As an alternative, we confirmed the expression of CDH13 in TECs by immunofluorescence (Figure 3e).
13. Line 182: The authors make an interesting observation: the transcription level of a few genes appear to be higher in adult TEC when compared to foetal and neonatal cells. It would be helpful to know whether increased levels of transcripts correlate with heightened protein expression and in which pathways these gene products are likely to be involved so as to explain their putative role in thymus involution and extend their observation beyond a mere correlation to time. As these observations concern the so-called immature TEC, it would again be instructive to know the nature of the cells’ developmental trajectory.
To gain more insights into the nature of immature TECs, we sub-clustered them and analyzed them at a higher resolution. We also used the data from a recently published study by Park. et al (Science 367, 868, 2020) to further validate our results. The results are presented in Figures 3a-d, Supplementary Figure 3a, and Supplementary Table 3). In addition, we used RNA Velocity to look at developmental trajectories and have included the data in Figure 4a.
14. Line 189 and Figure 3D: The statement that K15+ TEC are positioned at the cortico-medullary

junction is not supported by the data shown in the figure and thus does not provide the spatial correlate observed in the mouse for precursor cells that give rise to mTEC.

The immunofluorescence analysis suggests that there are more than one population of K15+ TECs: a population of K8+/K5+ cells with lower expression of K15 present at the cortico-medullary junction, likely representing immature TECs, and cells with higher levels of K15 expression present throughout the medulla, likely marking mTEC lo. Whether the low expressing cells represent progenitors that give rise to mTECs is unclear, as stated in the text.

15. Line 196 and following: The expression of ASCL1 is an interesting observation. It would be important for the authors to demonstrate that this gene product is not expressed as part of an AIRE-independent promiscuous gene expression programme in cTEC and mTEC and that known ASCL1 target genes are co-expressed in ASCL1+ TEC.

A figure demonstrating expression of known ASCL1 targets in TECs has been added (Supplementary Figure 3d). The expression of ASCL1 is also much broader than tissue specific antigens like insulin, supporting the idea that it is not expressed as part of a promiscuous gene expression program.

16. Line 217 - 221: Are the authors in a position to comment on the labelling of neuroendocrine cells, their frequency and the change in labelled cells over time.

Unfortunately, we are not able to address this point since our lineage tracing experiments did not include markers for neuroendocrine cells.

17. Line 246: Could the authors explain how they selected which genes from each pathway to display? Are there any measures of significance that can be attached to particular pathways in subtypes of TEC?

Genes were selected from the list of differentially expressed genes generated using the `tl.rank_genes_groups` function of SCANPY. They were selected based on their established role in each pathway. The p values for each gene are included in Supplementary Table 2.

18. Line 269: The paragraph's conclusion is an overstatement of the data shown and known in the literature and should thus be edited to better represent what is shown as novel data.

We have modified the language to 'Although it is not clear which subset of mTECs depends on this signaling pathway, our data identified corneocyte-like/post-AIRE mTECs, which have the highest levels of p53 activity, as an interesting candidate'.

19. Line 276: Details should be provided on what basis and by which computational means the re-clustering was achieved.

Details on how the sub-clustering was done has been added to the 'Single-cell RNA-seq and computational analysis' section of the Methods.

20. Lines 278-280: These novel epithelial populations are interesting. How certain are the authors that these cells are genuinely epithelial in nature and in the case of ionocytes indeed separate from Tuft cells and not just a mere difference in cell state? An immunofluorescent validation of these cells within thymic sections would be further helpful to characterize these cells. This would seem to be a sensible approach to validating these new populations.

All the cells that were subclustered as epithelial cells were part of one of the three epithelial clusters identified using known marker genes (Figure 1d-e and Supplementary Figure 1b). They expressed common epithelial markers such as EPCAM, CDH1 (E-cadherin) and at least one of the cytokeratin commonly used to identify TECs (KRT8 or KRT5). As for the differences between tuft cells and ionocytes, it is difficult to say if they are separate cell types or different cell states. They clustered together in our analysis, thus indicating a high degree of similarity. However, by increasing the resolution, we were able to identify a set of genes that are differentially expressed between the two subsets (see point #21 below). We have also further validated the presence of ionocytes in the medulla using a combination of CFTR and TRPM2 antibodies (Figure 5c).

Could the authors comment on why this population drops out completely in the post-natal sample?

?? It is not clear which cell population the reviewer is referring to. Both tuft cells and ionocytes are present in postnatal samples.

21. Line 286: It would be informative to show the differential gene expression between thymic ionocytes and Tuft cells and between these populations and their respective counterparts situated in the respiratory tract and the GI tract. This information would also provide further verification regarding the nature of these cells.

The differential gene expression between thymic ionocytes and tuft cells as well as a comparison with their respective counterparts in the respiratory tract (data from this study DOI: <https://doi.org/10.1038/s41467-020-16239-z>) is now provided in Supplementary Figure 5e-g and Supplementary Table 5.

22. Lines 324-326: Given the relatively shallow sequencing depth here, could the authors comment on how many TSA are detected overall and in each individual TEC subpopulation? How many AIRE-induced genes are driving the TSA score?

These questions are really hard to address since the SCANPY gene score function output is the average expression of all the genes on the list used to calculate the score. It doesn't tell us how many genes and exactly which TSAs are detected in each cluster. We are therefore not able to provide this information.

23. Line 342: This is interesting data, although I am surprised that much can be seen with such shallowing sequencing depth. Could the authors please include cTEC to demonstrate a lack of TSA expression in these cells too? How are the authors identifying some of the more infrequently expressed TSA as being specific to individual TEC subtypes? Are the reported TSA enrichments associated with a significant differential expression?

Plots showing low TSA and APS-1 scores in cTECs have been added to Supplementary Figure 6b-c. Unfortunately, this type of analysis doesn't allow us to determine in which TEC subtypes the infrequently expressed TSAs are found but instead gives an overall score for each cell. Since TSAs are typically expressed in very few cells, most of them are not found when analyzing differential expression.

24. Lines 348-351: How sure are the authors that the myoid cells are not utilising these genes for other cellular functions and are truly capable of acting as antigen-presenting cells? This has been a controversy in the literature and could potentially be at least partially answered by close interrogation of the single cell RNA-seq dataset presented here.

Our hypothesis is that myoid cells are likely not acting as antigen-presenting cells since they don't express high levels of HLA class I and II but that they serve as a source of antigens that can be picked up by thymic APCs. The text has been edited to clarify this point.

25. What is the median number of genes per cell detected overall and by cluster? This would be important in enabling critical interpretation of the deeper phenotyping of TSA presented here.

A violin plot of the number of genes per cell in each cluster has been added to Supplementary Figure 6a.

26. Have the authors considered using RNA-based lineage tracing analyses, such as RNA Velocity, to identify differentiation trajectories within their data as suggested in several remarks made above?

As suggested by the reviewer, we used RNA Velocity to analyze differentiation trajectories. The results are presented in Figure 4a.

Reviewer #3 (Transcriptome analyses, systems immunology) (Remarks to the Author):

The authors of this manuscripts have employed a single cell transcriptomics approach for the characterization of human thymic epithelial cell populations. They examined using this unbiased approach the composition of tissues from subjects corresponding to different developmental/age groups. Validation experiments were also conducted as follow on to some of the most original

findings.

The manuscript is well written and well presented, which makes the description of the work and findings easy to follow. The findings follow in the footsteps of earlier studies employing scRNAseq to investigate Thymic T-cell development, but those either addressed primarily T-cell development or focused on embryonic tissues. Most of those studies are discussed in this work. One notable exception being the recent paper of Voboril et al in Nat. Commun. (PMID: 32398640). It is not directly “competing” with their effort but it might still be worth discussing. The authors should also be commended for making their data public via the NCBI Gene Expression Omnibus.

We thank the reviewer for the positive comments. A reference to the paper suggested by the reviewer has been added. We also included a panel showing the expression of Toll-like receptors in TEC subsets in Figure 4f.

The only significant weakness stems from the fact that data and conclusions presented are based on profiles generated from at most two subjects per age group, and in the case on adults from only one subject. The fact that they examine something as fundamental as cell populations, without the need to quantify changes in relative abundance of these populations means that doing so may be adequate. However it remains somewhat unsettling to be drawing such general conclusions based on a single individual. This limitations should at least be acknowledged and discussed.

We agree with the reviewer and used the data from the Park et al. manuscript (Science 367, 868, 2020) to increase our sample size and validate our findings. The results are presented in Supplementary Figures 2-5. We verified that functional TECs were also reduced in their adult samples (Supplementary Figure 2g) and confirmed increased expression of a subset of immature TEC genes over time (Supplementary Figure 3a). The presence of tuft cells, ionocytes, and ciliated cells was also validated in their dataset (Supplementary Figure 5c-d), thus supporting the conclusions from our study.

It is otherwise excellent work. Here are a couple of additional minor comments.

- In the abstract, Line 36, the authors refer to “TEC” without giving a definition for the abbreviation. The text has been edited to add the definition.

- Line 185: “the expression of KRT15 in the thymus has not been reported before “ It is mentioned among a number differentially expressed in Thymic Epithelial Progenitors (although its significance was not highlighted investigated)

: <https://www.sciencedirect.com/science/article/pii/S2213671114001167>

We added the citation to the text.

- Generally, the authors have not discussed any limitations of the study (and specifically regarding the point mentioned above).

We agree with the reviewer that the low number of donors was a limitation of the study but believe that using the Park et al. dataset to validate our findings addresses this issue.

REVIEWERS' COMMENTS

Reviewer #1 (Remarks to the Author):

This is well revised, and I am supportive.

Reviewer #2 (Remarks to the Author):

The authors have satisfactorily answered the reviewer's queries.

Reviewer #3 (Remarks to the Author):

The concerns that have been raised earlier were well addressed in the revised manuscript.